# Thiazolidin-2,4-Dione Scaffold: An Insight into Recent Advances as Antimicrobial, Antioxidant, and Hypoglycemic Agents

**DOI:** 10.3390/molecules27196763

**Published:** 2022-10-10

**Authors:** Harsh Kumar, Navidha Aggarwal, Minakshi Gupta Marwaha, Aakash Deep, Hitesh Chopra, Mohammed M. Matin, Arpita Roy, Talha Bin Emran, Yugal Kishore Mohanta, Ramzan Ahmed, Tapan Kumar Mohanta, Muthupandian Saravanan, Rakesh Kumar Marwaha, Ahmed Al-Harrasi

**Affiliations:** 1Department of Pharmaceutical Sciences, Maharshi Dayanand University, Rohtak 124001, India; 2MM College of Pharmacy, Maharishi Markandeshwar (Deemed to be University), Mullana 133207, India; 3Department of Pharmaceutical Sciences, Sat Priya College of Pharmacy, Rohtak 124001, India; 4Department of Pharmaceutical Sciences, Chaudhary Bansi Lal University, Bhiwani 127021, India; 5College of Pharmacy, Chitkara University, Punjab 140401, India; 6Bioorganic and Medicinal Chemistry Laboratory, Department of Chemistry, Faculty of Science, University of Chittagong, Chittagong 4331, Bangladesh; 7Department of Biotechnology, School of Engineering & Technology, Sharda University, Greater Noida 201310, India; 8Department of Pharmacy, BGC Trust University Bangladesh, Chittagong 4381, Bangladesh; 9Department of Pharmacy, Faculty of Allied Health Sciences, Daffodil International University, Dhaka 1207, Bangladesh; 10Department of Applied Biology, School of Biological Sciences, University of Science and Technology Meghalaya, Ri-Bhoi 793101, India; 11Natural and Medical Sciences Research Centre, University of Nizwa, Nizwa 616, Oman; 12AMR and Nanotherapeutics Laboratory, Department of Pharmacology, Saveetha Dental College, Saveetha Institute of Medical and Technical Sciences (SIMATS), Chennai 600077, India

**Keywords:** thiazolidin-2,4-dione derivatives, antimicrobial activity, antioxidant activity, antihyperglycemic activity, patents granted, mechanism of action

## Abstract

Heterocyclic compounds containing nitrogen and sulfur, especially those in the thiazole family, have generated special interest in terms of their synthetic chemistry, which is attributable to their ubiquitous existence in pharmacologically dynamic natural products and also as overwhelmingly powerful agrochemicals and pharmaceuticals. The thiazolidin-2,4-dione (TZD) moiety plays a central role in the biological functioning of several essential molecules. The availability of substitutions at the third and fifth positions of the Thiazolidin-2,4-dione (TZD) scaffold makes it a highly utilized and versatile moiety that exhibits a wide range of biological activities. TZD analogues exhibit their hypoglycemic activity by improving insulin resistance through PPAR-γ receptor activation, their antimicrobial action by inhibiting cytoplasmic Mur ligases, and their antioxidant action by scavenging reactive oxygen species (ROS). In this manuscript, an effort has been made to review the research on TZD derivatives as potential antimicrobial, antioxidant, and antihyperglycemic agents from the period from 2010 to the present date, along with their molecular mechanisms and the information on patents granted to TZD analogues.

## 1. Introduction

Small heterocyclic structures containing nitrogen and sulfur have been under investigation for a long time owing to their therapeutic relevance. They offer a wide range of structural varieties and also possess a proven range of diversified therapeutic potentials. Amongst the extensive variety of heterocyclic scaffolds explored in the search for potent bioactive molecules in the process of drug discovery, the thiazolidin-2,4-dione (TZD) (Figure 1) ring system has been acknowledged as a significant scaffold in medicinal chemistry [1].

Thiazolidin-2,4-dione (Figure 1), also called glitazone, is a heterocyclic moiety that consists of a five-membered saturated thiazolidine ring with sulfur at 1 and nitrogen at 3, along with two carbonyl functional groups at the 2 and 4 positions. Substitutions of various moieties are possible only at the third and fifth positions of the Thiazolidin-2,4-dione (TZD) scaffold. The analogues of TZD offer a wide range of structural varieties [2] and also possess a proven range of diversified therapeutic potentials, such as antidiabetic [3,4,5], analgesic, anti-inflammatory [6,7,8], wound healing [9], antiproliferative [10,11,12,13,14], antimalarial [15], antitubercular [16,17], hypolipidemic [18], antiviral [19], antimicrobial, antifungal [20,21,22,23], and antioxidant properties [24,25], etc.

## 2. The History of Glitazones as Antidiabetics

TZDs are primarily used as hypoglycemic agents over a long time. Ciglitazone is the prototype of the TZD class, which was developed by Takeda Pharmaceuticals (Japan) in the early 1980s but has never been used as a medication due to its hepatotoxicity. In the year 1988, another TZD analog, named troglitazone, was developed by the Sankyo Company (Japan) as an antidiabetic agent, but it was also banned in the year 2000 due to its hepatic toxicity. In 1999, Pfizer (USA) and Takeda (Japan) jointly developed two molecules, pioglitazone (patented in 2002), and englitazone, and subsequently, Pfizer and Smithkline also developed rosiglitazone and darglitazone in the same year. Englitazone and darglitazone were discontinued due to their hepatotoxicity. However, pioglitazone was reported to be safe for use in the hepatic system and is currently in use as an antidiabetic agent. In 2001, rosiglitazone was reported to cause heart failure due to fluid retention and, hence, the Food and Drug Administration (FDA) restricted its use in the year 2010. However, in the year 2013, the restriction was removed by the FDA after a series of trials proved that there was no effect on the heart due to fluid retention. Another drug, lobeglitazone, was developed by Chong Kun Dang Pharmaceuticals (Korea) in 2013, which was approved by the Ministry of Food and Drug Safety of Korea. Some other molecules, such as netaglitazone, rivoglitazone, and balaglitazone, were also developed but then withdrawn during the clinical trial due to their severe toxicities, and these never came to the market. The structures of various TZDs are shown in Figure 2 [26,27]. 

## 3. Mechanism of Action

TZDs produce their biological response by stimulating the PPARγ receptor (antidiabetic activity) and cytoplasmic Mur ligase enzyme (antimicrobial activity), and by scavenging ROS (antioxidant activity). 

Family of PPARs:

PPARs are a subfamily of transducer proteins that act as transcriptional factors, belonging to the nuclear superfamily of retinoic acid receptors (RARs)/steroid receptors/thyroid hormone receptors (TRs), which are involved in different processes and also help in the regulation and expression of various genes that are vital for glucose and lipid metabolism [28,29]. These nuclear receptors were first identified in rodents in the year 1990s [26]. 

Isoforms of PPARs:

To date, three different types of PPARs have been identified, i.e., PPAR-α (NR1C1), PPAR-β/δ (NR1C2), and PPAR-γ (NR1C3), which are encoded by different genes.

Structure and biological functions of PPARs:

All of the PPAR isoforms have similar functional and structural features. The structure of PPAR has four principal functional domains, known as A/B, C, D, and E/F (Figure 3). The domain A/B, located at the NH_2_-terminal, comprises ligand-independent activation function 1 (AF-1) and causes the phosphorylation of PPAR. The domain C, also called the DNA-binding domain (DBD), has two zinc atoms that promote the binding of the peroxisome proliferator response element (PPRE) to PPAR in the target gene promoter region. The D site functions as a docking domain for the binding of the coactivators/corepressors to the DNA receptors. The E/F domain near the AF-2 region, also called the ligand-binding domain (LBD), is used to specify the ligand and also activates the binding of PPAR to PPRE, thereby promoting the targeted gene expression. The ligand-dependent activation function 2 (AF-2) recruits the PPAR co-factors that assist in the gene transcription processes. Their biological functions and distribution in the tissues, along with their agonists, are shown in Table 1 [26,30,31,32].

### 3.1. Mechanism of the TZD as an Antidiabetic

PPAR acts by either transactivation or by trans-repression to enact its antidiabetic activity. In transactivation, PPAR is activated upon binding with the exogenous ligand (TZD) or endogenous ligands (PGs, fatty acids, etc.). PPAR then heterodimerizes with the retinoid X factor (RXR) to form the PPAR–RXR complex. This complex binds with peroxisome proliferator response elements (PPRE) in the target gene with a coactivator that has histone acetylase activity and promotes the transcription of different genes involved in the cellular differentiation and glucose and lipid metabolism (Figure 4) [26,28]. In trans-repression, PPARs interact negatively with other signal transduction pathways, such as the nuclear factor kappa beta (NFκB) pathway, which controls various genes involved in inflammation and also regulates inflammatory mediators, such as leukocytes and cytokines, etc. (Figure 5) [28].

Thiazolidinediones (TZDs) are the most important synthetic moieties with PPAR-γ activation properties, which improve insulin resistance and, hence, lower blood glucose levels in type-II diabetes. In adipose tissues, TZDs activate PPAR-γ, which causes lipid uptake and the storage of triglycerides (TGs). White adipose tissues (WAT) then take up free fatty acids (FFAs) from the tissues (skeletal muscle, liver), where their growth obstructs insulin signaling, known as the lipid steal hypothesis. PPAR-γ also mediates the production of adipocytes. In macrophages, TZDs directly activate PPAR-γ, causing an anti-inflammatory M2 phenotype which leads to a decreased macrophage infiltration in the WAT. TZDs also help to reduce fibrosis and inflammation by acting on PPAR-γ in Kupffer and stellate cells, as well as the parenchymal cells of the steatosis liver, and also mediate atherosclerosis by interacting with PPAR-γ in the macrophages (Figure 6) [26,32,33].

### 3.2. Mechanism of TZD as Antimicrobial Agent

For bacterial viability, the bacterial cell wall is an important component in maintaining the cell shape and protection. Peptidoglycan is one of the major constituents of the bacterial cell wall and is found on the outer wall of cytoplasmic membrane. Its biosynthetic enzyme inhibition can lead to cell death. The enzymes involved in the synthesis can either be membrane-bound extracellular enzymes (penicillin-binding proteins) or cytoplasmic enzymes (Mur enzymes). Peptidoglycan peptide stem biosynthesis involves four ATP-dependent enzymes, known as the Mur ligases (Mur C-F). They mediate the formation of UDP-MurNAc-pentapeptide through the stepwise additions of MurC (L-alanine), MurD (D-glutamic acid), a diamino acid which is generally a meso-diaminopimelic acid, or MurE (L-lysine) and MurF (dipeptide D-Ala-D-Ala) to the D-lactoyl group of UDP-N-acetylmuramic acid (Figure 7). TZD molecules are supposed to inhibit these cytoplasmic ligases and, hence, lead to bacterial cell death [34,35,36,37].

### 3.3. Mechanism of TZD as an Antioxidant

Oxidative stress occurs due to the production of free radicals. Free radicals are chemically active molecules which are either deficient or have a greater number of electrons. Free radicals containing oxygen are the most significant free radicals, also known as reactive oxygen species (ROS). Oxidants activate various relevant enzymes, such as SOD, catalase, and NADPH oxidase, to convert the oxygen to ROS. ROS scavenge body cells in order to capture or donate protons, thus causing cell, proteins, and DNA damage. TZD derivatives are supposed to work by preventing the cascade effect produced through ROS propagation by donating its proton to the ROS (Figure 8) [38,39,40].

## 4. Biological Potential of the Thiazolidin-2,4-dione Analogues

### 4.1. Antimicrobial Activity

In the 1940s, the introduction of antibiotics was believed to have eliminated the curse of all infectious diseases. However, the irrational overuse of antibiotics has led to the development of resistance against antibiotics of several bacterial strains, which has surfaced as one of the serious public health issues. Some of these resistant strains, such as vancomycin-resistant enterococci (VRE) and multidrug-resistant Staphylococcus aureus (MRSA), can survive despite the presence of the majority of the antibiotics currently in use [20]. The appearance of multidrug-resistant microbial pathogens that are resistant to currently available antibiotics/antimicrobials has led to the urgent need for new chemical entities for the treatment of microbial diseases with a unique mechanism of action [41].

In the search of new antimicrobial agents, Shaikh et al. synthesized different series of (E)-5-(substitutedbenzylidene)thiazolidine-2,4-dione and their (6-thiocyanatobenzo- thiazol-2-yl)acetamide derivatives. The in vitro antimicrobial activity of derived molecules against various bacterial and fungal strains was evaluated using a broth microdilution assay measuring the minimum inhibitory concentration (MIC). Among the synthesized derivatives, compounds **am1**, **am2,** and **am3** were found to be effective against *E. coli*, *S. aureus,* and *C. albicans*, respectively, using ampicillin and griseofulvin as standards (Table 2, Figure 9) [42].

To recognize potential new compounds that are active against microbes, 5-benzylidene-2-4-thiazolidinedione-based antibacterial agents were developed by Zvarec et al. The synthesized derivatives were assessed for their in vitro antibacterial activity. Compounds **am4** and **am5** showed moderate antibacterial activity against *B. subtilis* (Table 3, Figure 9) [43].

Gaonkar et al. synthesized a new series of *N*-substituted thiazolidine-2,4-dione derivatives and screened for their in vitro antimicrobial potential against bacterial and fungal strains, using DMF as a solvent in the agar plate disc diffusion method, taking ciprofloxacin and ciclopiroxolamine as standard drugs. The result of the antimicrobial evaluation revealed that compounds **am6** and **am7** showed a superior antimicrobial potential (Table 4, Figure 9) [44].

Niwale et al. also developed various thiazolidin-2,4-dione analogues and evaluated them for their in vitro antibacterial potential by the serial tube dilution method using streptomycin as a reference drug. The results of the antimicrobial evaluation studies revealed that compounds **am8** and **am9** were active against all the tested strains of bacteria (Table 5, Figure 9) [45]. 

A new series of thiazolidine-2,4-diones containing a 1,2,3-triazole scaffold were synthesized by Sindhu et al. and evaluated for their in vitro antimicrobial potential against two bacterial and two fungal strains, using the disc diffusion and poisoned food methods, respectively. The results of the antimicrobial evaluation revealed that compounds **am10** and **am11** had superior antibacterial and antifungal activities in comparison to ciprofloxacin and fluconazole as the standard drugs (Table 6, Figure 9) [46].

Alagawadi et al. derived a sequence of TZD analogues containing the 1,3,4-thiadiazole moiety and screened them for their in vitro antimicrobial activity. The results of the antimicrobial evaluation studies showed compounds **am12** and **am13** to be active against all the tested strains of microbes, using ampicillin and ketoconazole as standards (Table 7, Figure 9 and Figure 10) [47].

In an attempt to identify prospective new TZD derivatives that are active against microbes, a new series of n-phenyl-acetamide derivatives of TZD were developed by Juddhawala et al. The synthesized products were further analyzed for their in vitro antibacterial potential by the disc diffusion method, using ciprofloxacin as the standard. The results of the antimicrobial evaluation revealed that compounds **am14** and **am15** had a better antimicrobial potential (Table 8, Figure 10) [48].

Patel et al. derived a series of thiazolidine-2,4-diones molecules containing the N-methoxybenzoyl substitution, and they were screened for their in vitro antimicrobial potential by the broth microdilution method, using ampicillin/griseofulvin as the standard, against different bacterial/fungal strains. The results revealed that compounds (**am16**) and (**am17**) had better antibacterial and antifungal activities (Table 9, Figure 10) [49].

Malik et al. derived a series of six new N-substituted 2, 4- thiazolidinedione compounds with active pharmacophores and screened them for their in vitro antibacterial potential against different bacterial strains, using the cup plate method, taking ciprofloxacin as the standard. The compounds (**am18**) and (**am19**) were found to be better antibacterial derivatives in the study (Table 10, Figure 10) [50].

Parekh et al. synthesized various TZD molecules containing a benzene sulfonamide scaffold and evaluated them for their in vitro antibacterial activity against different bacterial strains, using the disk diffusion method and taking ciprofloxacin as the standard drug. Among the derivatives tested, compounds **am20**, **am21**, **am22c,** and **am23** exhibited promising potential against the microbial strains and showed an activity comparable with that of the standard drug. (Table 11, Figure 10) [51]. 

Desai et al. synthesized a series of thiazolidine-2,4-diones with pyrrole and pyrazole rings. The minimum inhibitory concentration (MIC) of the synthesized derivatives was evaluated through the broth microdilution process and compared with two commercial antibiotics (ampicillin and griseofulvin). Among the derived molecules, compounds, **am24**, **am25**, **am26,** and **am27** were found to be the best antimicrobial molecules (Table 12, Figure 10) [52]. 

A new series comprising nine novel derivatives of thiazolidine-2,4-diones with a pyrazole ring were developed by Prakash et al. and screened for their in vitro antimicrobial potential against various fungal and bacterial strains, using the agar well diffusion method and poisoned food method, respectively, and they were compared with standard drugs ciprofloxacin and fluconazole. Compounds **am28** and **am30** were found to be promising antibacterial agents, while compound**s am29** and **am31** were found to be effective antifungal agents (Table 13, Figure 11) [53].

Prakash et al. synthesized another series of thiazolidine-2,4-diones molecules and screened them for their in vitro antimicrobial potential against various bacterial and fungal strains. Among the synthesized derivatives, compounds, **am32** and **am33** were found to be equipotent antibacterial and antifungal agents in comparison to the standard drugs ciprofloxacin and fluconazole as the antibacterial and antifungal agents, respectively (Table 14, Figure 11) [54].

To identify potential novel agents possessing antimicrobial activity, certain analogues of imidazolyl thiazolidin-2,4-dione and 5-substituted 2,4-thiazolidinedione were derived by Moorthy et al. and evaluated for their in vitro antimicrobial potential by measuring the MIC values using the agar streak dilution method. Compounds **am34**, **am35,** and **am36** showed moderate antimicrobial activity in comparison to the standard drugs, ciprofloxacin and ketoconazole (Table 15, Figure 11) [20].

Lobo et al. synthesized a series of 3,5-disubstituted thiazolidine-2,4-diones and screened them for their in vitro antimicrobial potential against different bacterial/fungal strains, using the cup plate method and taking ciprofloxacin and fluconazole as the standards. Among the synthesized analogues, molecules **am37** and **am38** exhibited a decent activity against all the tested bacterial and fungal strains compared with the standard drug used (Table 16, Figure 12) [55].

A series of substituted 5-(aminomethylene)thiazolidine-2,4-diones derivatives w active pharmacophores were derived by Mohanty et al. and tested for their in vitro antimicrobial potential against different strains of bacteria and fungi using the two-fold serial dilution method and poisoned food method, respectively. Compounds **am39** and **am40** showed a moderate antibacterial activity, while the compounds **am39**, **am40**, **am41,** and **am42** showed a promising antifungal activity compared to the standard drugs, ciprofloxacin and fluconazole respectively (Table 17, Figure 12) [22].

Alegaon et al. derived various TZD benzoic acid derivatives using a microwave irradiation procedure and evaluated their antimicrobial potential against different bacterial and fungal species. Compounds **am43** and **am44** were found to be the most active antimicrobial agents in comparison to the standard drugs, ampicillin, ciprofloxacin, and ketoconazole (Table 18, Figure 12) [56].

To improve the previously recognized lead structure, a series of TZD-5-acetic acid amides were developed by Alegaon et al. and screened for their in vitro antimicrobial activity against various fungal and bacterial strains using the twofold sequential dilution procedure, taking commercial drugs, i.e., ciprofloxacin, ampicillin, and ketoconazole, as the standards. The compound **am45** was found to possess the best antimicrobial potential amongst all the derived derivatives (Table 19, Figure 12) [57]. 

Avupati et al. synthesized various ((oxoprop-1-enyl)benzylidene)-1,3-thiazolidine-2,4-dione derivatives and tested them for their in vitro antimicrobial potential against many bacterial and fungal strains using the agar well diffusion method. Compounds **am46** and **am47** showed promising antimicrobial activity when compared with the standard drugs, chloramphenicol and ketoconazole (Table 20 and Table 21, Figure 12) [58].

Liu et al. reported a new series of Thiazolidin-2,4-dione containing methyl benzoic acid molecules and screened them for their in vitro antibacterial potential against many bacterial strains using a serial dilution method with a 96-well microtiter plate. Norfloxacin and ofloxacin were taken as the standards. The results of the antibacterial activity study showed that all the prepared derivatives were inactive against the Gram-negative bacterial strains but active against the Gram-positive bacterial strains. The compounds **am48** and **am49** exhibited a good antibacterial potential in comparison to the standard drugs amongst all the synthesized molecules (Table 22, Figure 12 and Figure 13) [59].

Alegaon et al. synthesized another series of TZD acetic acid derivatives and evaluated them for their in vitro antibacterial potential against different microbial strains using a two-fold serial dilution method, taking ciprofloxacin and ketoconazole as the standards. The results of the antimicrobial activity evaluation revealed that the compounds **am50** and **am51** possessed a moderate antimicrobial activity (Table 23, Figure 13) [60]. 

Da Silva et al. synthesized 5-arylidene-thiazolidine-2,4-dione derivatives and screened them against different microbial strains to determine their MICs and MBCs (minimum bactericidal concentrations) using the 96-well plate method. The results revealed that compounds **am52** and **am53** possessed excellent antimicrobial activity against the tested strains in comparison to the standard drug, cefalexin (Table 24, Figure 13) [61].

Rekha et al. synthesized 5-arylidene-thiazolidine-2,4-dione analogues and tested them against different microbial strains by measuring their zone of inhibition using a slightly modified cup plate method. The results revealed that compound **am54** provided a better inhibition of *B. subtilis* and *S. aureus*, while **am55** possessed excellent potency against *P. vulgaris* when compared with the standard drug, Amoxycillin (Table 25, Figure 13) [62].

Nastasa et al. synthesized a few 2,4-thiazolidinedione derivatives containing a chromene scaffold and tested them for their in vitro antimicrobial potential against certain bacterial and fungal species using the disk diffusion method. The results of the antimicrobial evaluation revealed that compounds **am56** and **am57** were more active than the commercial drugs gentamicin and fluconazole against the tested species of the microbial strains (Table 26, Figure 13) [21].

A series of 3-aryl-5-arylidine thiazolidine-2, 4-dione molecules were derived by Purohit et al. and the MICs of the derived compounds were obtained using a twofold serial tube dilution technique, taking ciprofloxacin, norfloxacin, griseofulvin, and fluconazole as the standards. The screening results revealed that compounds **am58** and **am59** possess a comparable antimicrobial potency to the standard drugs (Table 27, Figure 13) [63].

Mangasuli et al. derived various coumarin-thiazolidinone derivatives and evaluated their antimicrobial potential against different bacterial and fungal species. Compounds **am60** and **am61** were found to be the most active antimicrobial agents in comparison to the standard drugs, ciprofloxacin and ketoconazole (Table 28, Figure 13) [64].

Alhameed et al. derived various thiazolidine-2,4-dione carboxamide and amino acid derivatives and evaluated their antimicrobial potential against different bacterial and fungal species, taking imipenem and fluconazole as the reference drugs. Compound **am62** was found to be the most active analogue against *S. aureus* (Table 29, Figure 13) [65].

### 4.2. Hypoglycemic Activity

Diabetes has been recognized as a metabolic syndrome characterized by raised blood sugar (or blood glucose) levels, affecting a large number of people, both male and female, across the globe. Type-II diabetes mellitus (T2DM) is a multifaceted disease that often affects longevity/the life span due to grave damage to the eyes, kidneys, heart, blood vessels, and nerves [3,66]. According to statistical data, around 200 million people are suffering from diabetes to date, and this figure will be increased to 350 million by 2025 and more than 360 million by 2030 [67,68]. The derivatives of thiazolidine-2,4-done, viz., ciglitazone, troglitazone, rosiglitazone, and many more, were approved as antidiabetic agents but were withdrawn from the market due to their severe hepatotoxicity [69]. Therefore, there is a pressing need to synthesize newer hypoglycemic agents without toxicity.

In the search for new hypoglycemic agents, Jawale et al. synthesized various 1-((2,4-dioxothiazolidin-5-yl)methyl)-3-(substitutedbenzenesulfonyl)urea derivatives and screened them for their in vivo hypoglycemic activity in Sprague–Dawley strain male albino rats using the sucrose-loaded model (SLM) with commercially available metformin as standard. The results of the antihyperglycemic activity test revealed that some of the synthesized compounds in the sucrose-loaded rat model showed substantial antihyperglycemic activity. Furthermore, compounds **ad1**, **ad2,** and **ad3** showed better activity in the series (Table 30, Figure 14) [70]. 

Mishra et al. synthesized thiazolidine-2,4-dione derivatives with the *N*-chromen-3-yl ethyl moiety and screened them for their in vivo antidiabetic activity in male Wistar rats weighing 150–250 gm, using the alloxan-induced diabetic model and taking pioglitazone as a standard. The results of the antidiabetic screening demonstrated that all the synthesized derivatives had a promising glucose-reducing potential. Furthermore, compounds **ad4**, **ad5,** and **ad6** were found to be almost equipotent with the standard (Table 31, Figure 14) [66].

A new series of TZD-acrylic acid alkyl ester molecules were derived by Kumar et al. and screened for their in vivo hypoglycemic potential in neonatal Wistar male rats using the streptozotocin-induced diabetic model. The results of the antihyperglycemic activity test revealed that all the compounds possessed a moderate plasma glucose level (%PGL)-reducing activity in comparison to the marketed drug rosiglitazone. Furthermore, compounds **ad7**, **ad8,** and **ad9** possessed the maximum activity in the series (Table 32, Figure 14) [71].

Gautam et al. designed a series of substituted 5-(arylidene)-thiazolidine-2,4-diones and evaluated them as potential aldose reductase inhibitors by the molecular docking technique (PDB id: 1AH3) using Molegro Virtual Docker (MVD 2012.5.5.0), along with the clinically available drug epalrestat. The affinity of the designed compounds for the aldose reductase enzyme was measured by calculating the Mol-dock score. The results of the docking studies showed that all the compounds had similar or slightly better activity than the clinical drug epalrestat. Compound **ad10** in the series possessed the maximum activity and can be used as a lead structure (Table 33, Figure 14) [68].

Swathi et al. synthesized certain 5-substitutedaryl/heteroarylmethylidene-1,3-thiazolidine-2,4-diones and observed their hypoglycemic activity using in silico studies, viz., protein binding energy calculation, drug-likeness, and docking simulation studies, etc. The docking studies were carried out on the PPARγ enzyme (2PRG protein), obtained from a protein data bank, taking rosiglitazone as a standard drug. The results of the docking studies indicated that the synthesized compounds were equipotent with or more potent than standard. Compounds **ad11**, **ad12,** and **ad13** exhibited the maximum potential in the series (Table 34, Figure 14) [72].

Evcimen et al. synthesized thiazolyl-2,4-thiazolidinediones derivatives and screened them for their in vivo hypoglycemic potential by determining their aldose reductase inhibition using male Wistar albino rats. A Kidney enzyme was isolated and NADPH oxidation was calculated spectrophotometrically using D-L-glyceraldehyde as a substrate. The results of the study revealed that some of the synthesized derivatives possessed significant inhibitory activity. Furthermore, compound **ad14** possessed the maximum inhibitory potential in the series (Table 35, Figure 14) [73].

Datar et al. synthesized 3,5-disubstituted thiazolidin-2.4-dione molecules and screened them for their in vivo antidiabetic potential using a sucrose-loaded model tested on Wistar rats. The results of the activity test revealed that all the derivatives possessed mild to good activity as compared to the marketed drug, pioglitazone. Compounds **ad15** and **ad16** were equipotent with standard (Table 36, Figure 14 and Figure 15) [69].

Verma et al. synthesized compounds **ad17** and **ad18** (Figure 15) containing indolyl- linked benzylidene molecules with a thiazolidine-2,4-dione scaffold and observed their in silico antidiabetic activity using the Surflex-dock module. The docking was carried out on 1I7I protein complexed with tesaglitazar obtained from the protein data bank using human peroxisome-proliferator-activated receptor gamma (PPARγ), and the compounds were found more potent than standard. The results are depicted in Table 37 [74].

Nazreen et al. synthesized thiazolidine-2,4-dione derivatives containing the 1,3,4-oxadiazole moiety and screened them for their in vitro PPARγ transactivation assay and in vivo antidiabetic potential using a streptozocin-induced diabetic rat model, comparing them with the commercial drugs rosiglitazone and pioglitazone. Most of the synthesized derivatives possessed mild to moderate activity in PPARγ transactivation and were equipotent with the standards or had a greater potential in lowering the blood glucose levels. Furthermore, compounds **ad19** and **ad20** were found to be most active derivatives in the series, without hepatotoxicity (Table 38, Figure 15) [75].

Avupati et al. synthesized various (*Z*)-5-(4-((*E*)-3-(substituted)-3-oxoprop-1-enyl)benzylidene)-1,3-thiazolidine-2,4-dione derivatives and evaluated them for their in vivo antidiabetic activity using the streptozotocin-induced diabetic model tested on Wistar albino rats, and the results were compared using the marketed drug rosiglitazone. The results of the study revealed that compounds **ad21** and **ad22** were promising antidiabetic derivatives (Table 39, Figure 15) [58].

Wang et al. synthesized various 5-(benzylidene)thiazolidine-2,4-dione derivatives and evaluated them as competitive inhibitors of protein tyrosine phosphatase 1B (PTP1B), using p-nitrophenyl phosphate (pNPP) as a substrate and ursolic acid as a positive control. The results of activity study revealed that compounds **ad23** and **ad24** were the most active PTP1B inhibitors amongst all the synthesized molecules (Table 40, Figure 15) [76].

Swapna et al. derived different 5-[4′-(S]ubstituted)sulphonylbenzylidene]-2,4-thiazolidinedione molecules and screened them for their in vivo antidiabetic activity using an alloxan-induced tail tipping diabetic model tested on Wistar albino rats. All the synthesized derivatives produced comparable results to the standard drug metformin. Compounds **ad25**, **ad26,** and **ad27** exhibited the maximum activity (Table 41, Figure 15) [77].

Kumar et al. derived a series of TZD derivatives with indole at the 5th position and screened them for their in vivo antidiabetic activity using an alloxan-induced tail tipping diabetic model of Wistar albino rats and compared them with the standard drug glibenclamide. Amongst all the synthesized derivatives, molecules **ad28**, **ad29** and **ad30** exhibited significant antidiabetic activity (Table 42, Figure 15) [78].

Mahapatra et al. derived a series of (Z)-3,5-disubstituted thiazolidine-2,4-dione derivatives containing the thiophen moiety and screened them for their in vitro protein tyrosine phosphatase 1B (PTP1B) inhibition activity colorimetrically, using suramin as a standard inhibitor. The results of the assay revealed that most of the synthesized derivatives were weak PTP1B inhibitors. However, compound **ad31** exhibited an almost equipotent inhibition with that of standard inhibitor (Table 43, Figure 16) [79].

To further improve the previously recognized lead analogue; a series of (Z)-4-((2,4-dioxothiazolidin-5-ylidene)methyl)phenyl-substitutedbenzenesulfonates were derived by Mahapatra^b^ et al. and screened for their in vitro protein tyrosine phosphatase 1B (PTP1B) inhibition activity, taking the commercial drug suramin as the standard. Compounds **ad32** and **ad33** were found to be the most potent in the series and possessed greater inhibitory activity than the standard (Table 44, Figure 16) [4].

Badiger et al. derived a series of thiazolidine-2,4-diones derivatives with a 1,3,4-thiadiazole scaffold and screened them for their in vivo antidiabetic activity using an alloxan-induced tail tipping diabetic model of Wistar albino rats in comparison to the standard drug pioglitazone. Amongst all the synthesized derivatives, molecules **ad34** and **ad35** exhibited significant activity (Table 45, Figure 16) [80].

Alam et al. derived a new series of N-substituted-methyl-1,3-thiazolidine-2,4-dione molecules and evaluated them for their in vivo antidiabetic potential inn Wistar rats using the streptozotocin-induced diabetic model. The results of antidiabetic evaluation revealed that compounds **ad36**, **ad37**, **ad38,** and **ad39** were the most active antidiabetic compounds, with equipotent activity to the standard drug glibenclamide (Table 46, Figure 16) [81].

Patil et al. synthesized a new series of TZD molecules and evaluated their in vivo antidiabetic potential in Wistar rats using an alloxan-induced tail tipping diabetic model, taking metformin and pioglitazone as standard drugs. The results of the antidiabetic evaluation revealed that compounds **ad40** and **ad41** were the most active antidiabetic compounds (Table 47, Figure 16) [82].

Garg et al. derived a series of 5-(substituted)arylidene-3-substituted-benzylthiazolidin-2,4-dione analogues and screened them for their in vivo antidiabetic activity using an alloxan-induced tail tipping diabetic model tested on Wistar albino rats, and the results were compared with the commercial drug rosiglitazone. Amongst all the synthesized derivatives, **ad42** and **ad43** exhibited significant activity (Table 48, Figure 16) [83].

In the search for a lead compound a series of thiazolidine-2,4-dione derivatives clubbed with azole moiety was derived by Senthilkumar et al. and screened for their in vitro α-amylase and α-glucosidase inhibition potential, taking the commercial drug acarbose as the standard. Compounds **ad44, ad45,** and **ad46** were found to be most potent amongst all the derived derivatives and possessed an almost equipotent inhibitory activity with the standard (Table 49, Figure 16 and Figure 17) [84].

Nikalje et al. derived a series of thiazolidine-2,4-dione acetamide derivatives and screened them for their in vivo antidiabetic activity using an alloxan-induced tail tipping diabetic model of Wistar albino rats, taking pioglitazone as a standard drug. Amongst all the synthesized derivatives, molecules **ad47, ad48**, and **ad49** exhibited a promising activity (Table 50, Figure 17) [85].

Sucheta et al. derived a series of 5-(aryl/alkyl)benzylidene-thiazolidine-2,4-dione derivatives and screened them for their in vitro α-amylase inhibitory activity, taking commercial drug acarbose as the standard. Compounds **ad50** and **ad51** were found to be the most potent of the derived derivatives and exhibited good inhibitory activity as compared to standard (Table 51, Figure 17) [5].

Srikanth et al. derived a series of thiazolidine-2,4-dione derivatives with a quinolin-8-yloxy]benzyl scaffold and screened them for their in vivo antidiabetic activity using an alloxan-induced tail tipping diabetic model tested on Wistar albino rats, taking rosiglitazone as the standard drug. Amongst all the synthesized derivatives, molecules **ad52, ad53**, and **ad54** exhibited moderate activity (Table 52, Figure 17) [86].

Kadium et al. synthesized a series of (z)-5(4-hydroxy-3-methoxybenzylidine)-2,4- thiazolidinedione analogues and tested them for their in vivo antidiabetic potential using a sucrose-loaded model tested on Wistar rats. The results of the activity study revealed that all the derivatives possessed good activity, comparable with the marketed drug pioglitazone. Compounds **ad55** and **ad56** were found to be more potent than standard (Table 53, Figure 17) [87]. 

### 4.3. Antioxidant Activity

Antioxidants are the molecules used to delay or prevent the oxidation of oxidizable substrate when used in low concentrations. The key function of antioxidants is to prevent any cellular damage caused by free radicals by reacting with and stabilizing them. Free radicals are not only produced by smoke, obesity-inducing foods, cigarette, radiation, or exposure to frequent chemical constituents (lead, polycyclic aromatic hydrocarbon, cadmium, etc.), but are also formed by normal cellular processes, during several biochemical reactions, and in diseases such as tumors, atherosclerosis, diabetic and cardiac diseases, etc. Antioxidants work by preventing the cascade effect produced by reactive oxygen species (ROS) propagation by donating their proton to the ROS. Antioxidants neutralize free radicals, hence preventing them from attacking the cell. Natural antioxidants act as natural cleansers and body detoxifiers. They convert toxins of the body into harmless waste products [38,39]. Therefore, the synthesis of highly potent new antioxidant agents is required.

For the development of new antioxidant compounds, Kumar et al. synthesized various thiazolidine-2,4-dione derivatives containing 5-substituted aryl/alkyl moieties, and their in vitro antioxidant potential was evaluated using an 2, 2-diphenyl-1-picrylhydrazyl (DPPH) radical scavenging method, taking ascorbic acid as the reference drug. The results of the antioxidant activity test revealed that all the synthesized compounds exhibited substantial antioxidant activity. Furthermore, compounds **ao1** and **ao2** were found to be the most potent derivatives in the series (Table 54, Figure 18) [24].

Sucheta et al. derived a series of 5-(substituted)benzylidene-thiazolidine-2,4-dione derivatives, and in vitro*,* antioxidant studies were performed, applying a DPPH free radical scavenging method and taking ascorbic acid as a reference molecule. Compounds **ao3** and **ao4** were found to be the most potent among all the derivatives and possessed a good antioxidant activity (Table 55, Figure 18) [5].

Mishra et al. synthesized substituted 5-benzylidene-3-(2-oxo-2-(2-oxo-2H-chromen-3-yl)ethyl)thiazolidine-2,4-dione derivatives and screened them in in vitro antioxidant studies, applying the 2, 2-diphenyl-1-picrylhydrazyl (DPPH) radical scavenging method and using ascorbic acid as a standard. The results of the assay revealed that compounds **ao5** and **ao6** possessed a better activity than the other derivatives in the series (Table 56, Figure 18) [66].

Rekha et al. synthesized 5-arylidene-thiazolidine-2,4-dione molecules and tested them for their antioxidant potential applying the DPPH radical method. The results revealed that compounds **ao7** and **ao8** possessed excellent potency when compared with the standard drug, ascorbic acid (Table 57, Figure 18) [62].

A new series of (*E*)-5-(2-(5-(substitutedbenzylidene)-2,4-dioxothiazolidin-3-yl)acetyl)-2-hydroxybenzamide were derived by Marc et al. and screened them for their in vitro antioxidant potential using ABTS and DPPH radical scavenging assays. The results of antioxidant activity tests revealed that all the compounds exhibited moderate to potent radical scavenging activity in comparison with Trolox (6-hydroxy-2,5,7,8-tetramethylchroman-2-carboxylic acid), ascorbic acid, and butylated hydroxytoluene (BHT) as the standard compounds. Furthermore, compounds **ao9** and **ao10** exhibited even better activity than the reference compounds (Table 58, Figure 18 and Figure 19) [25].

Khare et al. designed a series of thiazolidine-2,4-dione scaffolds and tested them for their in vitro antioxidant potential using the DPPH radical scavenging method, and the results were compared with standard ascorbic acid. The results of the antioxidant activity test revealed that all the compounds showed moderate radical scavenging activity. Furthermore, compounds **ao11** and **ao12** possessed better activity than other compounds (Table 59, Figure 19) [38].

Koppireddi et al. synthesized various TZD acetamide analogues with thiazole and benzothiazole moieties and observed their antioxidant potential using superoxide anion scavenging activity, DPPH radical scavenging activity, lipid peroxidation inhibition, and erythrocyte hemolysis inhibition assays. Compounds **ao13** and **ao14** displayed decent activity when compared with standard ascorbic acid and luteolin (Table 60, Figure 19) [88].

Shahnaz et al. derived *N*-substituted methyl-2,4-thiazolidinediones molecules and tested them for their in vitro antioxidant potential using a DPPH radical scavenging assay, taking ascorbic acid as the standard. The results of the antioxidant activity tests revealed that all the compounds showed moderate radical scavenging activities. Furthermore, compounds **ao15** and **ao16** exhibited better activity than the other compounds in the series (Table 61, Figure 19) [89].

Nyaki et al. prepared ((1H-imidazol-2-yl)benzylidene)thiazolidin-2,4-dione molecules and tested them for their in vitro antioxidant potential using a DPPH radical scavenging assay, taking ascorbic acid as the standard. The results of antioxidant activity tests revealed that all the compounds showed moderate to good radical scavenging activities. Furthermore, compounds **ao17** and **ao18** exhibited better activity than the standard drug (Table 62, Figure 19) [90].

Sameeh et al. derived Benzo[d][1,3]dioxol-5-ylmethylene)-thiazolidin-2,4-dione analogues and screened them for their in vitro antioxidant potential using a DPPH radical scavenging assay, taking ascorbic acid as the reference drug. The results of the assay revealed that molecules **ao19** and **ao20** exhibited a better antioxidant potential than the standard compound (Table 63, Figure 19) [91].

Sameeh et al. again derived different thiazolidin-2,4-dione analogues and evaluated them for their in vitro antioxidant activity using a DPPH radical scavenging assay, taking ascorbic acid as the reference drug. The results of the assay revealed that molecules **ao21** and **ao22** exhibited a better antioxidant potential than other drugs in the series (Table 64, Figure 19) [92].

## 5. Patents Grant Information

Due to the significant potential of the TZD derivatives, many investigators working on thiazolidinediones analogues have contributed a new dimension to the drug design. Several patents have been granted to the TZD analogues, which are shown in Table 65.

## 6. Conclusions and Future Perspective

The TZD moiety plays a central role in the biological functioning of several essential molecules and possesses significant medicinal potential. The synthetic methodologies are simple and versatile; hence, the discovery of new TZD analogues is easily attainable. The substitution of various moieties at the third and fifth positions of the Thiazolidin-2,4-dione (TZD) scaffold offers the medicinal chemist the potential to derive novel TZD molecules. TZDs are primarily used as antidiabetic agents but also exhibit diverse therapeutic activities, such as antimicrobial, antiviral, anticancer, and antioxidant activities, etc. The present review article attempted to explore the pharmacological potential of TZDs for application as antimicrobial, antioxidant, and hypoglycemic agents, along with their mechanism of action. A brief description of the recent patents granted to TZDs possessing different biological activities was also provided. This review paper will not only be helpful to researchers working on the development of new TZD analogues based on medicinal chemistry, but also for designing new drug molecules in the future. Future investigations of TZD molecules using other heteroatoms may provide us with more encouraging results. Based on the results of this review, TZDs may be considered as an auspicious class of drugs that can be utilized to form new antidiabetic, antimicrobial, or antioxidant agents with a minimal toxic effect.

## Figures and Tables

**Figure 1 molecules-27-06763-f001:**
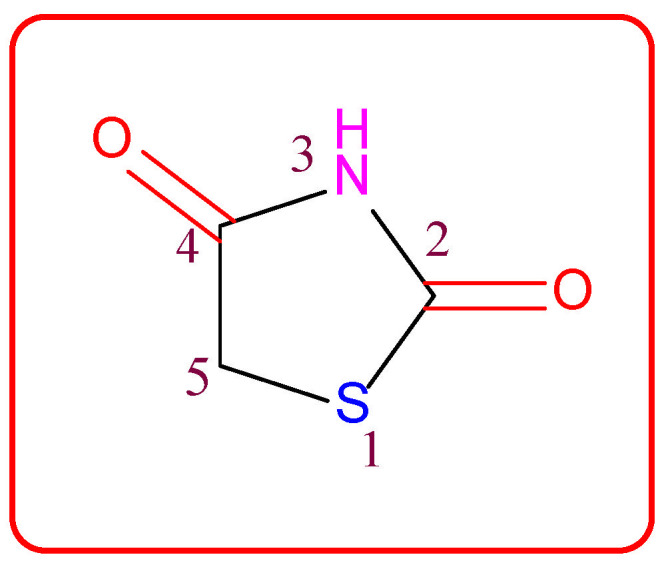
Structure of the thiazolidine-2,4-dione scaffold.

**Figure 2 molecules-27-06763-f002:**
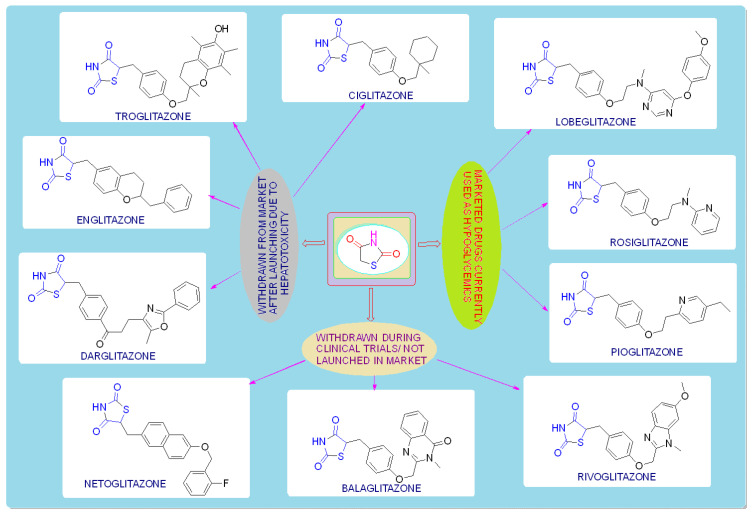
Antidiabetic molecules developed with the thiazolidin-2,4-dione moiety.

**Figure 3 molecules-27-06763-f003:**

Principal functional domains of PPARs.

**Figure 4 molecules-27-06763-f004:**
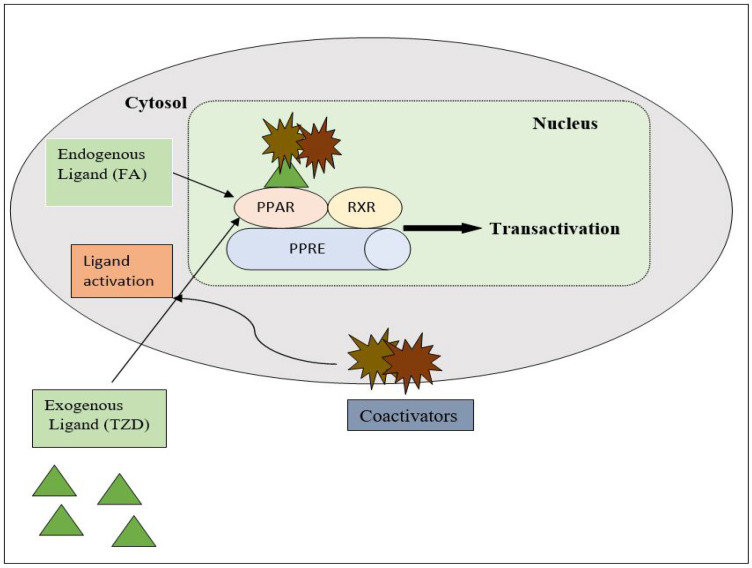
Gene transcription mechanisms of PPAR.

**Figure 5 molecules-27-06763-f005:**
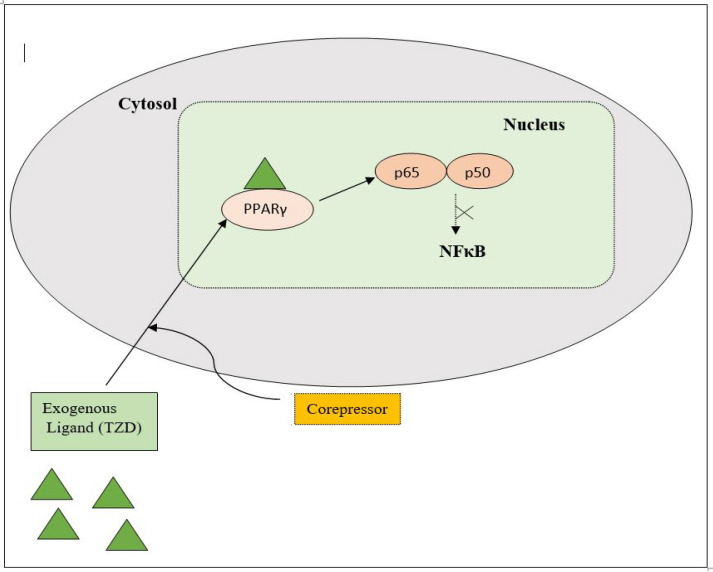
Gene trans-repression mechanisms of PPAR.

**Figure 6 molecules-27-06763-f006:**
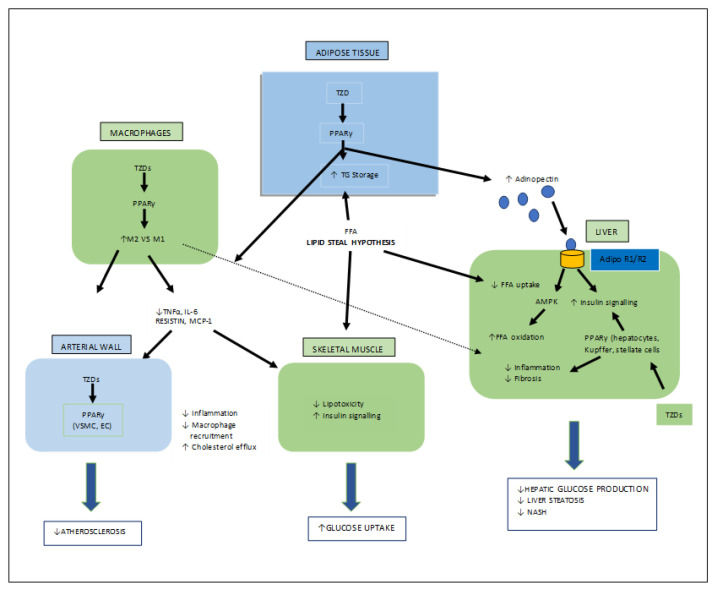
Various target organs/sites of TZD-PPARγ.

**Figure 7 molecules-27-06763-f007:**
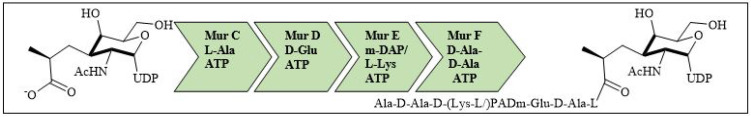
Peptidoglycan peptide stem formation by the Mur ligases enzymes.

**Figure 8 molecules-27-06763-f008:**
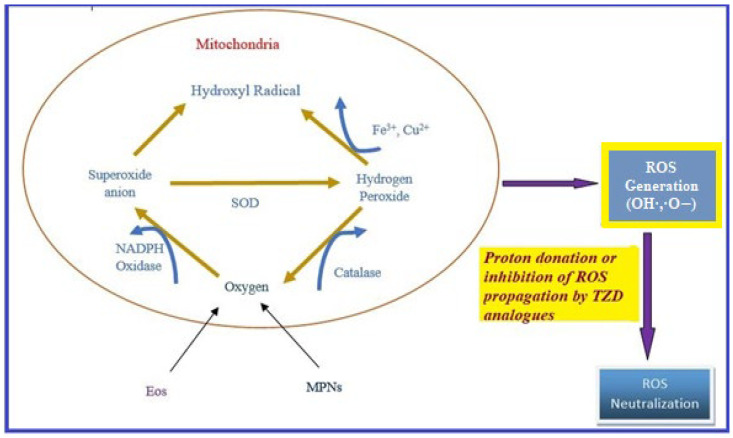
ROS generation and antioxidant scavenging mechanism of TZDs.

**Figure 9 molecules-27-06763-f009:**
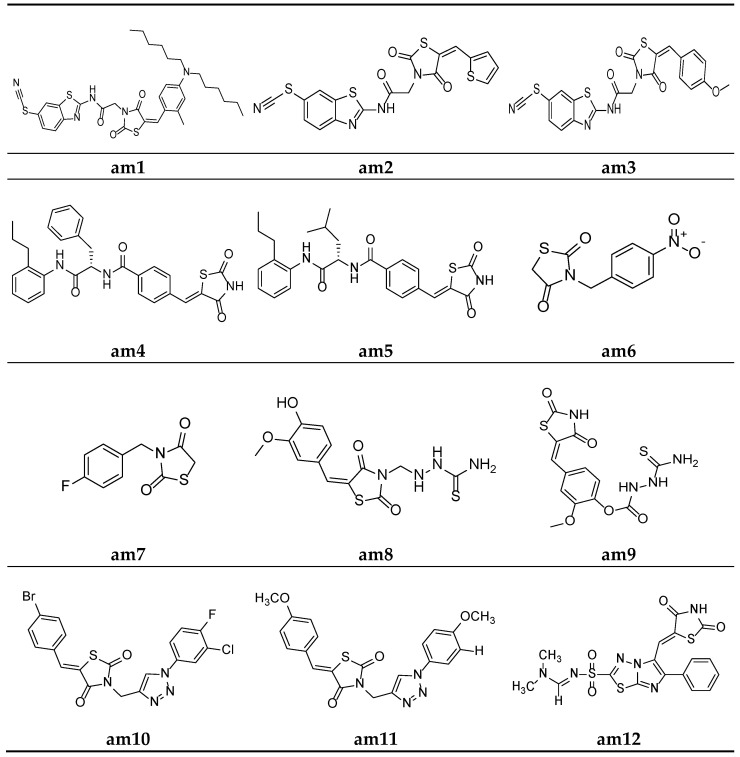
Molecular structures of the compounds (**am1**–**am12**).

**Figure 10 molecules-27-06763-f010:**
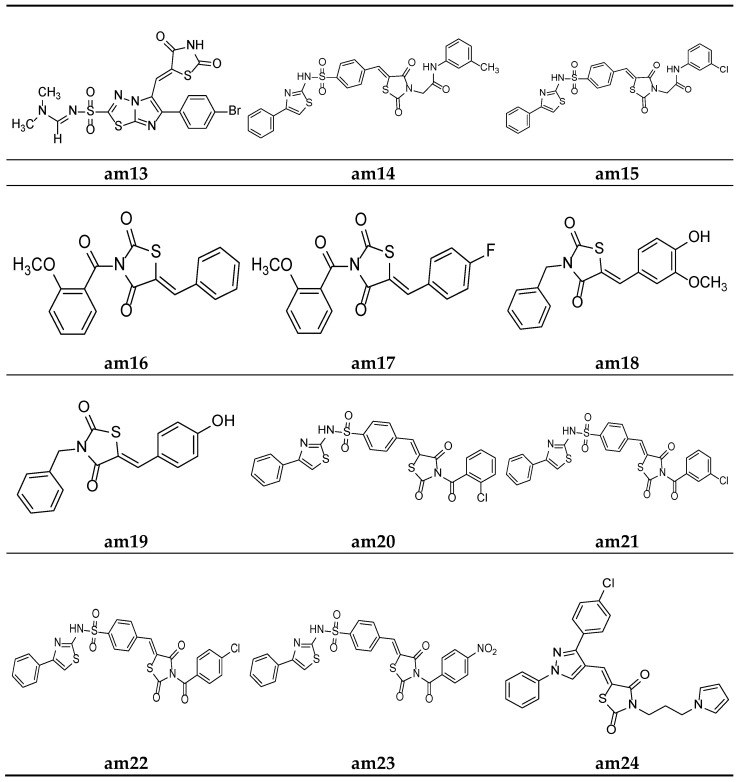
Molecular structures of the compounds (**am16**-**am24**).

**Figure 11 molecules-27-06763-f011:**
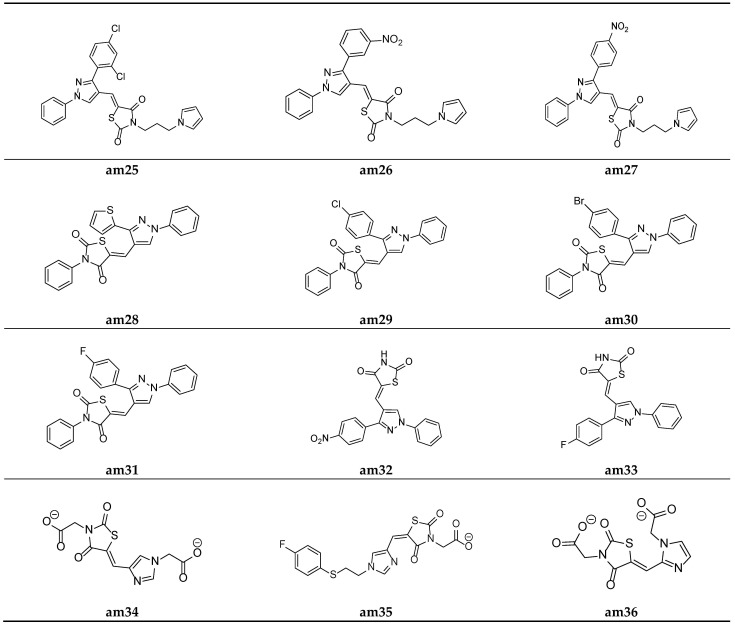
Molecular structures of the compounds **(am25**–**am36**).

**Figure 12 molecules-27-06763-f012:**
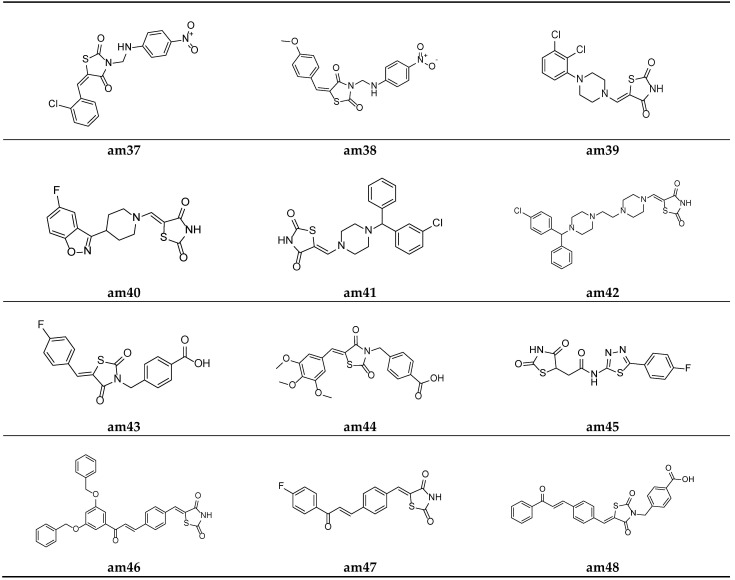
Molecular structures of the compounds (**am37**–**am48**).

**Figure 13 molecules-27-06763-f013:**
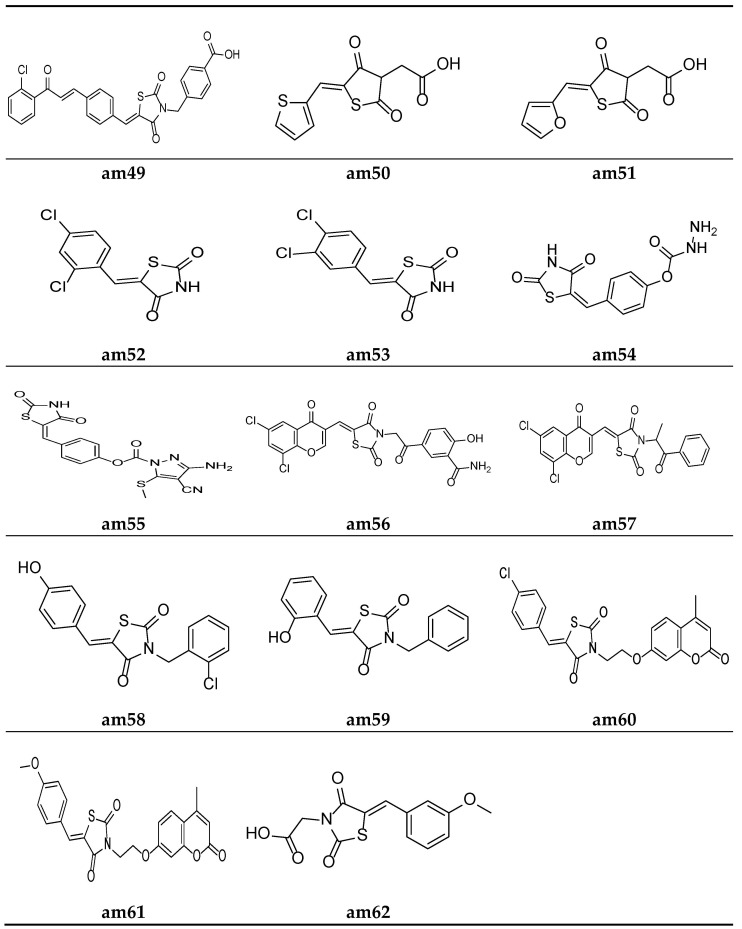
Molecular structures of the compounds (am49–am62).

**Figure 14 molecules-27-06763-f014:**
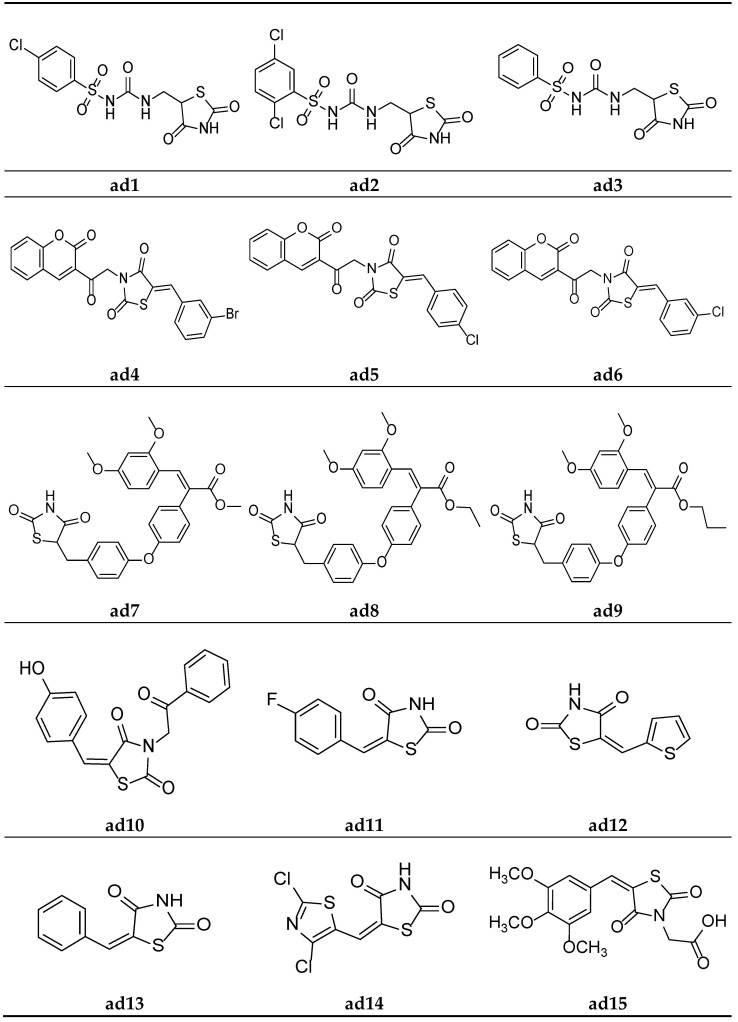
Molecular structures of the compounds (**ad1**–**ad15**).

**Figure 15 molecules-27-06763-f015:**
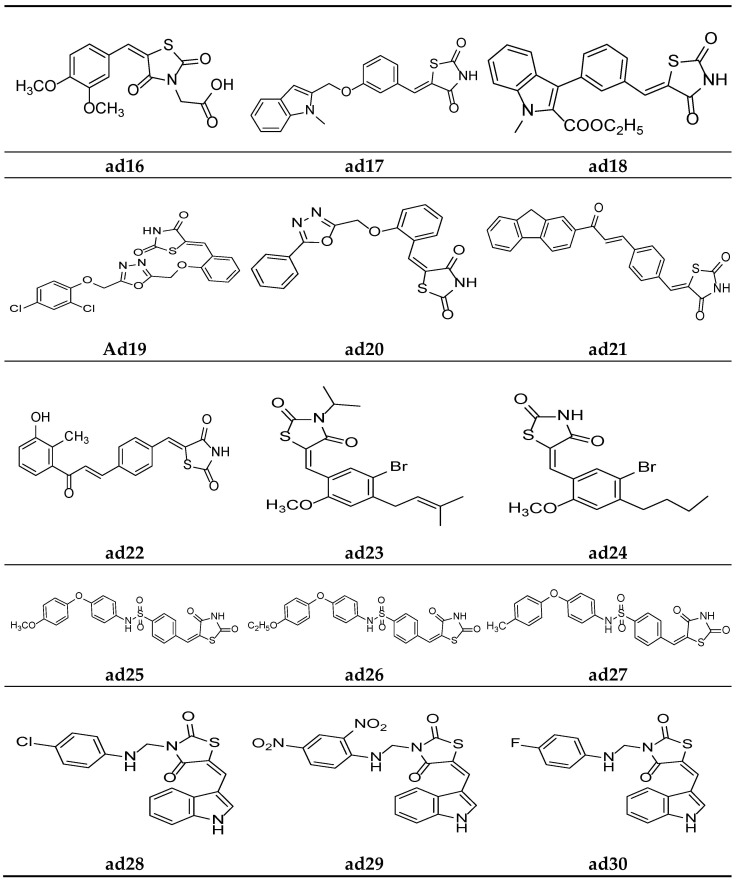
Molecular structures of the compounds (**ad16**–**ad30**).

**Figure 16 molecules-27-06763-f016:**
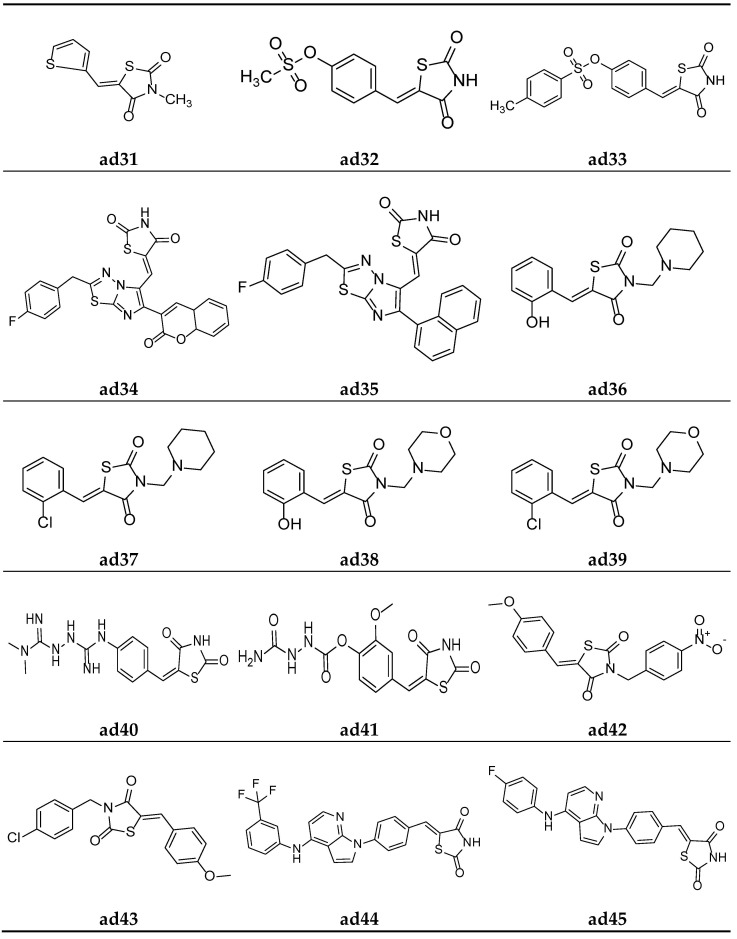
Molecular structures of the compounds (**ad31**–**ad45**).

**Figure 17 molecules-27-06763-f017:**
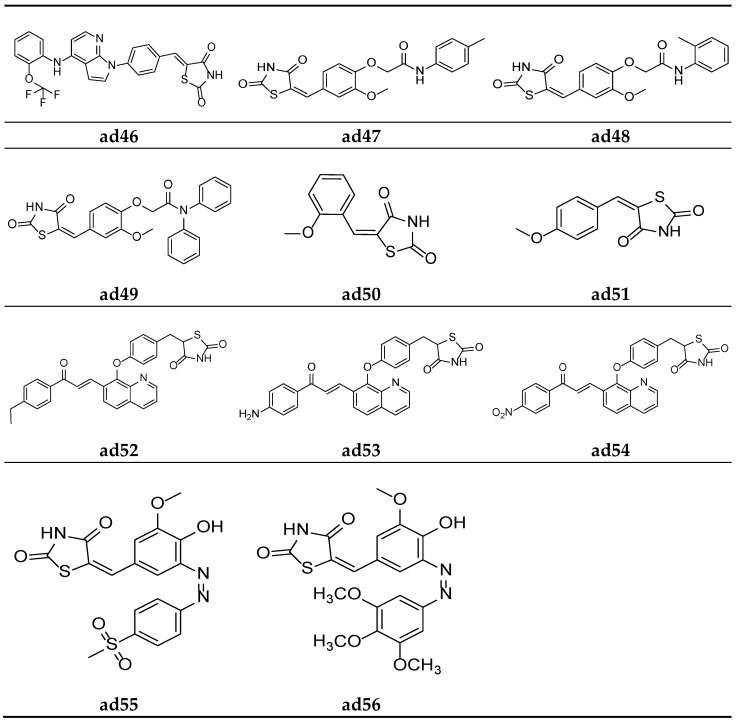
Molecular structures of the compounds (**ad46**–**ad56**).

**Figure 18 molecules-27-06763-f018:**
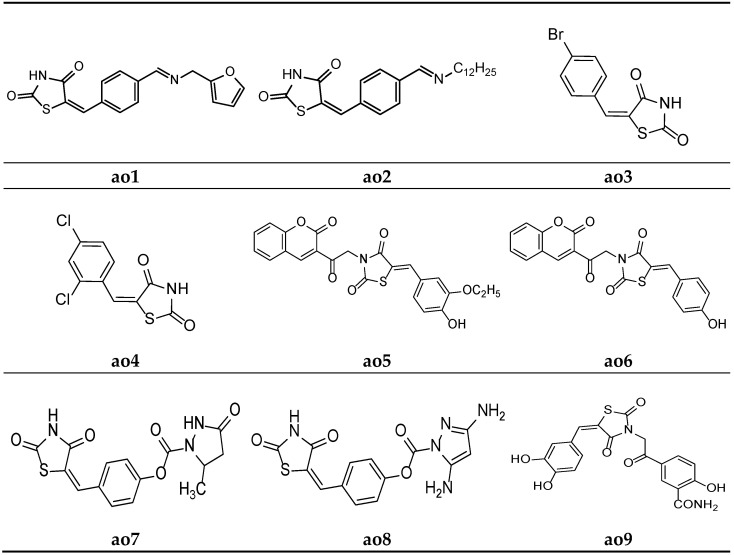
Molecular structures of the compounds (**ao1**–**ao9**).

**Figure 19 molecules-27-06763-f019:**
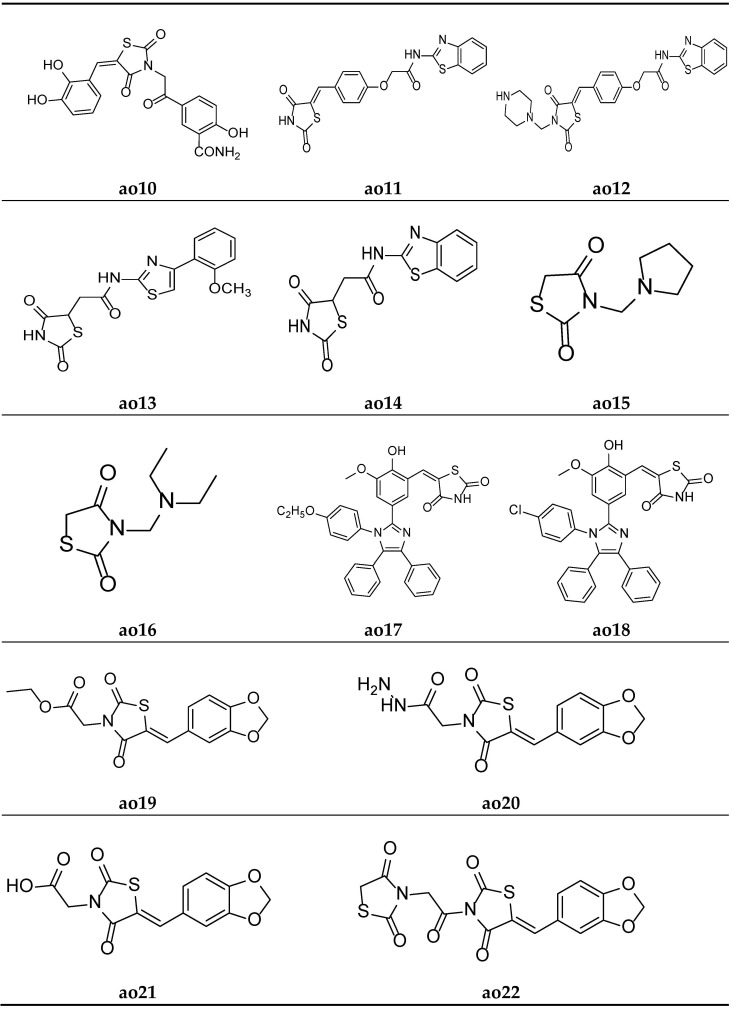
Molecular structures of the compounds (**ao10-ao22**).

**Table 1 molecules-27-06763-t001:** Biological functions of PPARs.

Isoform	Agonists	Location	Biological Functions
PPAR-γ	TZDs, unsaturated fatty acids (oleate, linoleate), arachidonic and eicosapentaenoic acids, and prostanoid.	Mainly in the brown and white adipose tissue and, to a lesser extent, in the placenta, colon mucosa, and immune cells (Peyer’s patches in the digestive tract, monocytes, and macrophages).	- Sensitization of insulin. - Adipocyte differentiation and adipogenesis. - Cellular growth and inflammation.
PPAR-α	fibrates (clofibrate, fenofibrate, and bezafibrate), unsaturated fatty acids, 8-(S) hydroxyl eicosatetraenoic acid, B4 leukotriene, prostaglandin E, or farnesol.	Kidney cortex, skeletal muscles, enterocytes, cardiomyocytes, and hepatocytes.	- Oxidation of fatty acids, mostly in the heart, liver, and muscles. - Reduces inflammation in both the liver and vascular wall. - Regulation of energy homeostasis.
PPAR-β/δ	Fatty acids	In almost all the tissues, especially in the brain, skin and, adipose tissue.	- Regulates fat oxidation. - Regulates the genes involved in adipogenesis. - Lipoprotein metabolism. - Glucose homeostasis. - Cholesterol metabolism. - Inflammation. - Atherosclerosis.

**Table 2 molecules-27-06763-t002:** Antimicrobial results of the compounds (**am1**–**am3**).

Compounds	Microbial Strains (MIC = µg/mL)						
	Bacterial Species				Fungal Species		
	*E. coli*	*P. aeruginosa*	*S. aureus*	*S. pyrogenes*	*C. albicans*	*A. niger*	*A. clavatus*
**am1**	50	250	500	500	250	500	500
**am2**	500	500	50	200	500	250	250
**am3**	100	200	250	500	100	1000	1000
Ampicillin	100	100	250	100	-	-	-
Griseofulvin	-	-	-	-	500	100	100

**Table 3 molecules-27-06763-t003:** Antimicrobial results of the compounds (**am4**–**am5**).

Compounds	Microbial Strains (MIC = µg/mL)		
	Bacterial Species		
	*S. aureus*	*S. epidermidis*	*B. subtilis*
**am4**	>64	>64	8
**am5**	>64	>64	8

**Table 4 molecules-27-06763-t004:** Antimicrobial results (inhibitory zone = mm diameter) of the compounds (**am6**–**am7**).

Compounds	Microbial Strains (Inhibitory Zone = mm Diameter)							
	Bacterial Species				Fungal Species			
	*S. aureus*	*E. coli*	*P. aeruginosa*	*K. pneumoniae*	*A. fumigatus*	*A. flavus*	*P. marneffei*	*C. albicans*
**am6**	15	19	20	16	18	15	17	19
**am7**	17	18	21	15	19	16	17	18
Ciprofloxacin	19	20	25	18	-	-	-	-
Ciclopiroxolamine	-	-	-	-	22	18	20	20

**Table 5 molecules-27-06763-t005:** Antimicrobial activity (MIC = µg/mL) of the compounds (**am8**–**am9**).

Compounds	Microbial Strains (MIC = µg/mL)		
	Bacterial Species		
	*S. aureus*	*P. aeruginosa*	*B. subtilis*
**am8**	31.25	31.25	31.25
**am9**	31.25	31.25	31.25
Streptomycin	3.90	3.90	3.90

**Table 6 molecules-27-06763-t006:** Antimicrobial activity of the compounds (**am10**–**am11**).

Comp.	Bacterial Species		Fungal Species	
	Inhibitory Zone = mm Diameter		Mycelial Growth Inhibition (%)	
	*S. aureus*	*P. aeruginosa*	*A. niger*	*A. flavus*
**am10**	21.6	22.3	35.5	43.3
**am11**	18.6	19.3	53.3	55.8
Ciprofloxacin	26.6	24.0	-	-
Fluconazole	-	-	81.1	77.7

**Table 7 molecules-27-06763-t007:** Antimicrobial activity (MIC = µg/mL) of the compounds (**am12**–**am13**).

Comp.	Microbial Strains (MIC = µg/mL)							
	Bacterial Species				Fungal Species			
	*S. aureus*	*E. coli*	*P. aeruginosa*	*E. faecalis*	*A. niger*	*A. flavus*	*C. neoformans*	*C. albicans*
**am12**	8	128	32	4	4	4	2	1
**am13**	8	64	64	8	4	4	8	4
Ampicillin	1	2	2	2	NT	NT	NT	NT
Ketoconazole	NT	NT	NT	NT	1	2	1	2

NT: not tested.

**Table 8 molecules-27-06763-t008:** Antimicrobial activity (Inhibitory Zone = mm diameter) of the compounds (**am14**–**am15**).

Comp.	Microbial Strains (Inhibitory Zone = mm Diameter)			
	Bacterial Species			
	*S. aureus*	*P. aeruginosa*	*B. subtilis*	*E. coli*
**am14**	22	18	24	19
**am15**	20	21	24	20
Ciprofloxacin	32	31	33	30

**Table 9 molecules-27-06763-t009:** Results of antimicrobial screening of the synthesized compounds (**am16**–**am17**).

Comp.	Microbial Strains (MIC = µM/mL)						
	Bacterial Species				Fungal Species		
	*E. coli*	*P. aeruginosa*	*S. aureus*	*S. pyrogenes*	*C. albicans*	*A. niger*	*A. clavatus*
**am16**	250	125	125	100	500	500	>1000
**am17**	500	62.5	250	500	200	500	500
Ampicillin	250	100	100	100	-	-	-
Griseofulvin	-	-	-	-	500	100	100

**Table 10 molecules-27-06763-t010:** Antimicrobial results (inhibitory zone = mm diameter) of the compounds (**am18**–**am19**).

Comp.	Microbial Strains (Inhibitory Zone = mm Diameter)	
	Bacterial Species	
	*E. coli*	*B. subtilis*
**am18**	16	13
**am19**	13	11
Ciprofloxacin	25	25

**Table 11 molecules-27-06763-t011:** Antimicrobial screening results (inhibitory zone = mm diameter) of the compounds (**am20**–**am23**).

Comp.	Microbial Strains (Inhibitory Zone = mm Diameter)			
	Bacterial Species			
	*S. aureus*	*P. aeruginosa*	*B. subtilis*	*E. coli*
**am20**	24	24	26	25
**am21**	26	24	25	24
**am22**	24	22	26	23
**am23**	25	24	24	24
Ciprofloxacin	32	31	33	30

**Table 12 molecules-27-06763-t012:** Results of antimicrobial screening of the synthesized compounds (**am24**–**am27**).

Comp.	Microbial Strains (MIC = µM/mL)						
	Bacterial Species				Fungal Species		
	*E. coli*	*P. aeruginosa*	*S. aureus*	*S. pyrogens*	*C. albicans*	*A. niger*	*A. clavatus*
am24	500 ± 4.01 *	50 ± 3.60	100 ± 1.00 **	50 ± 2.13 *	>1000	50 ± 2.20 *	500 ± 3.60 **
am25	50 ± 2.10	100 ± 3.05 ***	50 ± 3.21	50 ± 3.01	500 ± 1.30 **	25 ± 0.23 ***	50 ± 2.31
am26	50 ± 2.63 *	50 ± 3.21 **	100 ± 3.00 *	12.5 ± 2.56 **	500 ± 2.44	50 ± 4.33	100 ± 2.36 ***
am27	12.5 ± 2.05 **	100 ± 4.01 *	50 ± 1.33 ***	12.5 ± 3.44 *	500 ± 4.10 *	125 ± 3.12 **	50 ± 1.22 *
Ampicillin	250 ± 2.05	100 ± 0.98	100 ± 1.52	100 ± 2.06	-	-	-
Griseofulvin	-	-	-	-	500 ± 0.80	100 ± 1.98	100 ± 2.01

* *p* < 0.05 significant ** *p* < 0.01 moderately significant *** *p* < 0.001 extremely significant.

**Table 13 molecules-27-06763-t013:** Antimicrobial activity of the compounds **(am28**–**am31**).

Comp.	Bacterial Species		Fungal Species	
	Inhibitory Zone = mm Diameter		Mycelial growth Inhibition (%)	
	*S. aureus*	*B. subtilis*	*A. niger*	*A. flavus*
**am28**	22.3	21.6	50	33.3
**am29**	20.6	18.3	55.5	61.1
**am30**	21.6	20.6	38.8	55.5
**am31**	19.3	20.6	55.5	55.5
Ciprofloxin	26.0	24.0	-	-
Fluconazole	-	-	81.1	77.7

**Table 14 molecules-27-06763-t014:** Antimicrobial activity of the compounds **(am32**–**am33**).

Comp.	Bacterial Species		Fungal Species	
	Inhibitory Zone = mm Diameter		Mycelial Growth Inhibition (%)	
	*S. aureus*	*B. subtilis*	*A. niger*	*A. flavus*
**am32**	23.3	21.3	61.1	63.3
**am33**	20.6	21.6	82.5	78.8
Ciprofloxin	26.0	24.0	-	-
Fluconazole	-	-	81.1	77.7

**Table 15 molecules-27-06763-t015:** Results of antimicrobial screening of the synthesized compounds **(am34**–**am36**).

Comp.	Microbial Strains (MIC = µM/mL)					
	Bacterial Species				Fungal Species	
	*E. coli*	*P. aeruginosa*	*S. aureus*	*S. epidermidis*	*A. niger*	*A. fumigatus*
**am34**	1.6	0.56	1.9	1.4	8.8	2.3
**am35**	1.6	2.8	3.8	2.2	7.9	1.7
**am36**	3.2	1.4	2.7	3.39	8.2	3.4
Ciprofloxacin	0.2	0.25	0.39	0.2	-	-
Ketoconazole	-	-	-	-	6.1	0.23

**Table 16 molecules-27-06763-t016:** Results of antimicrobial activity of the synthesized compounds **(am37**–**am38**).

Comp.	Microbial Strains (Inhibitory Zone = mm Diameter)					
	Bacterial Species				Fungal Species	
	*E. coli*	*P. aeruginosa*	*S. aureus*	*B. subtilis*	*A. niger*	*C. albicans*
**am37**	28	24	28	26	25	27
**am38**	26	24	28	28	24	26
Ciprofloxacin	28	25	26	26	-	-
Fluconazole	-	-	-	-	25	26

**Table 17 molecules-27-06763-t017:** Results of antimicrobial screening of the synthesized compounds **(am39**–**am42**).

Comp.	Bacterial Species				Fungal Species	
	MIC = µg/mL				Mycelial Growth Inhibition (%)	
	*S. aureus*	*B. subtilis*	*E. coli*	*P. auroginosa*	*A. niger*	*A. flavus*
**am39**	16	16	16	16	60.0	60.4
**am40**	16	32	16	16	60.0	60.0
**am41**	16	16	64	32	61.2	61.2
**am42**	32	64	16	32	61.6	60.2
Ciprofloxacin	5	5	5	5	-	-
Fluconazole	-	-	-	-	75.3	74.6

**Table 18 molecules-27-06763-t018:** Antimicrobial activity (MIC = µg/mL) of the compounds **(am43**–**am44**).

Comp.	Microbial Strains (MIC = µg/mL)							
	Bacterial Species				Fungal Species			
	*S. aureus*	*E. coli*	*P. aeruginosa*	*E. faecalis*	*A. niger*	*A. flavus*	*C. neoformans*	*C. albicans*
**am43**	2	4	4	4	4	4	4	2
**am44**	8	8	8	8	4	4	4	2
Ampicillin	2	NT	NT	2	NT	NT	NT	NT
Ciprofloxacin	NT	2	2	NT	NT	NT	NT	NT
Ketoconazole	NT	NT	NT	NT	1	2	1	2

NT: not tested.

**Table 19 molecules-27-06763-t019:** Antimicrobial activity (MIC = µg/mL) of the compound **(am45**).

Comp.	Microbial Strains (MIC = µg/mL)							
	Bacterial Species				Fungal Species			
	*S. aureus*	*E. coli*	*P. aeruginosa*	*E. faecalis*	*A. niger*	*A. flavus*	*C. neoformans*	*C. albicans*
**am45**	4	8	8	4	4	8	8	8
Ampicillin	2	NT	NT	2	NT	NT	NT	NT
Ciprofloxacin	NT	2	2	NT	NT	NT	NT	NT
Ketoconazole	NT	NT	NT	NT	1	2	1	2

NT: not tested.

**Table 20 molecules-27-06763-t020:** Antibacterial activity (MIC = µg/mL) of the compounds **(am46**–**am47**).

Comp.	Microbial Strains (MIC = µg/mL)							
	Gram Positive Bacterial Species				Gram Negative Bacterial Species			
	*S. aureus*	*M. luteus*	*B. subtilis*	*B. pumilus*	*E. coli*	*P. aeruginosa*	*K. pneumonia*	*P. vulgaris*
**am46**	16	16	16	16	32	16	16	32
**am47**	32	32	16	16	16	16	16	32
Chloramphenicol	16	16	16	16	16	16	16	16

**Table 21 molecules-27-06763-t021:** Antifungal activity (MIC = µg/mL) of the compounds (**am46**–**am47**).

Comp.	Fungal Species			
	*C. albicans*	*A. niger*	*A. oryzae*	*P. chrysogenum*
**am46**	16	16	16	16
**am47**	32	32	32	16
Ketoconazole	16	16	16	16

**Table 22 molecules-27-06763-t022:** Inhibitory activity (MIC, µg/mL) of the compounds **am48** and **am49** against the bacteria and clinical isolates of multidrug-resistant Gram-positive strains.

Comp.	Microbial Strains (MIC = µg/mL)							
	Bacterial Species				Clinical Isolates of Gram Positive Bacterial Species			
	*S. aureus 4220*	*S. aureus 503*	*E. coli 1356*	*E. coli 1682*	*MRSA 3167*	*MRSA 3506*	*QRSA 3505*	*QRSA 3519*
**am48**	1	2	>64	>64	1	0.5	2	2
**am49**	1	2	>64	>64	1	1	2	2
Norfloxacin	2	2	16	16	8	4	>64	>64
Ofloxacin	1	2	>64	>64	>64	>64	1	1

**Table 23 molecules-27-06763-t023:** Antimicrobial activity (MIC = µg/mL) of the compounds (**am50**–**am51**).

Comp.	Microbial Strains (MIC = µg/mL)							
	Bacterial Species				Fungal Species			
	*S. aureus*	*E. coli*	*P. aeruginosa*	*E. faecalis*	*A. niger*	*A. flavus*	*C. neoformans*	*C. albicans*
**am50**	16	32	16	16	32	32	16	16
**am51**	16	64	32	16	16	32	64	16
Ciprofloxacin	2	2	2	2	NT	NT	NT	NT
Ketoconazole	NT	NT	NT	NT	1	2	1	2

NT: not tested.

**Table 24 molecules-27-06763-t024:** Results of antimicrobial screening of the synthesized compounds **(am52**–**am53**).

Comp.	Bacterial Species				
	MIC = µg/mL (MBC = µg/mL)				
	*B. subtilis*	*S. aureus*	*M. luteus*	*M. smegmatis*	*E. faecalis*
am52	2 (4)	8 (16)	2 (4)	2 (4)	16 (32)
am53	2 (4)	8 (16)	2 (4)	2 (4)	16 (32)
Cefalexin	<2 (2)	8 (16)	<2 (2)	>128 (>128)	>128 (>128)

**Table 25 molecules-27-06763-t025:** Antimicrobial screening results of the compounds (**am54**–**am55**).

Comp.	Microbial Strains			
	Inhibitory Zone = mm Diameter			
	*B. subtilis*	*S. aureus*	*E. coli*	*P. vulgaris*
**am54**	25 ± 0.23	20 ± 2.53	14 ± 1.23	12 ± 1.59
**am55**	14 ± 1.25	15 ± 0.94	15 ± 0.81	28 ± 1.23
Amoxycillin	28 ± 1.23	25 ± 1.34	17 ± 0.99	31 ± 0.41

**Table 26 molecules-27-06763-t026:** Antimicrobial activity of the compounds (**am56**–**am57**).

Comp. 10/5/1 (mg/mL)	Inhibitory Zone = mm Diameter				
	Microbial Species				
	*L. monocytogenes*	*S. aureus*	*S. typhi*	*E. coli*	*C. albicans*
**am56**	28/28/28	28/28/28	18/18/18	18/18/18	22/22/22
**am57**	22/22/20	24/28/28	20/18/16	18/18/16	18/18/16
Gentamicin	18	19	18	22	-
Fluconazole	-	-	-	-	28

**Table 27 molecules-27-06763-t027:** Antimicrobial activity (MIC = µg/mL) of the compounds (**am58**–**am59**).

Comp.	Microbial Strains (MIC = µg/mL)						
	Bacterial Species				Fungal +-		
	*S. aureus*	*E. faecalis*	*E. coli*	*K. pneumoniae*	*A. niger*	*A. flavus*	*C. albicans*
**am58**	1	1	62.5	62.5	4	2	4
**am59**	4	31.25	62.5	62.5	16	16	31.25
Norfloxacin	1	3.1	10	0.1	-	-	-
Ciprofloxacin	2	2	2	1	-	-	-
Griseofulvin	-	-	-	-	100	7.5	500
Fluconazole	-	-	-	-	8	8	16

**Table 28 molecules-27-06763-t028:** Antimicrobial activity (MIC = µg/mL) of the compounds (**am60**–**am61**).

Comp.	Microbial Strains (MIC = µg/mL)							
	Bacterial Species				Fungal Species			
	*S. aureus*	*B. subtilis*	*E. coli*	*P. aeruginosa*	*A. harzianum*	*A. flavus*	*P. chrysogenum*	*C. albicans*
**am60**	0.5	0.5	1	1	1	1	1	1
**am61**	4	4	2	2	8	8	8	2
Ciprofloxacin	2	2	2	2	NT	NT	NT	NT
Ketoconazole	NT	NT	NT	NT	2	2	2	2

NT: not tested.

**Table 29 molecules-27-06763-t029:** Antimicrobial activity of the compounds (**am62**).

Comp.	Inhibitory Zone = mm Diameter				
	Microbial Species				
	*B. subtilis*	*S. aureus*	*P. aeruginosa*	*E. coli*	*C. albicans*
**am62**	-	20	14	7	7
Imipenem	34	30	35	20	-
Fluconazole	-	-	-	-	28

**Table 30 molecules-27-06763-t030:** Hypoglycemic effect of the compounds (**ad1**–**ad3**) on sucrose-loaded hyperglycemic rats.

Comp.	Dose (mg/dL)	% Activity	Significance
**ad1**	100	17.2	*p* < 0.01
**ad2**	100	16.5	*p* < 0.01
**ad3**	100	15.8	*p* < 0.01
Metformin	100	27.0	*p* < 0.001

**Table 31 molecules-27-06763-t031:** Antidiabetic potential of the compounds (**ad4**–**ad6**) in alloxan-induced diabetic rats.

Comp.	Blood Glucose Level (mg/dL)				Log P	Change in Blood Glucose Level (from 1st to 6th Hour)
	0 h	1 h	3 h	6 h		
**ad4**	312 ± 12.78	270 ± 10.67	135 ± 9.89	127 ± 7.76	3.22	185
**ad5**	303 ± 4.89	255 ± 7.56	211 ± 2.98	133 ± 10.56	2.95	170
**ad6**	302 ± 5.98	273 ± 7.65	211 ± 7.34	137 ± 6.34	2.95	165
Pioglitazone	276 ± 4.84	213 ± 3.44	114 ± 5.49	89 ± 3.26	3.58	187

(N = 6) All data represent the mean ± SEM analyzed by one-way analysis of variance (ANOVA) using Dunnett’s multiple comparison test applied for statistical analysis.

**Table 32 molecules-27-06763-t032:** Hypoglycemic activity of the compounds (**ad7**–**ad9**) using a streptozotocin-induced diabetic rat model.

Comp.	% PGL Reduction
Control	0.433 ± 1.17
**ad7**	46.13 ± 4.96 ^a^
**ad8**	46.03 ± 3.08 ^a^
**ad9**	45.96 ± 4.51 ^a^

One-way ANOVA followed by Dunnett’s test. ^a^ *p* < 0.01. Values are presented as mean ± S.E.M. (N = 6).

**Table 33 molecules-27-06763-t033:** Hypoglycemic effect of the compound **ad10** depicted by docking score.

Comp.	Mol-Dock Score	Re-Rank Score	H-Bond Score
**ad10**	−141.292	−87.586	−7.001
Epalrestat	−113.889	−70.997	−1.481

**Table 34 molecules-27-06763-t034:** Antidiabetic potential of the compounds (**ad11**–**ad13**) using molecular docking studies.

Comp.	Docking Score (kcal/mol)	Interacting Residues
**ad11**	−10.49	His 323, His 449, Ser 289, Tyr 473
**ad12**	−10.12	His 323, His 449, Ser 289, Tyr 473
**ad13**	−10.04	His 323, His 449, Ser 289, Tyr 327
Rosiglitazone	−9.48	His 323, His 449, Cys 285, Tyr 473

**Table 35 molecules-27-06763-t035:** Aldose reductase inhibitory activity of the compound ad14.

Comp.	% Inhibition			IC_50_(M)
	10^−4^M	10^−5^M	10^−6^M	
**ad14**	91.11 ± 3.59	49.91 ± 2.49	14.06 ± 1.88	3.446 × 10^−5^ ± 0.30 × 10^−5^

**Table 36 molecules-27-06763-t036:** Antidiabetic potential of the compounds (**ad15**–**ad16**) using a sucrose-loaded model in rats.

Comp.	Blood Glucose Level (mg/dL)					% Reduction in Blood Glucose Level
	0 min	30 min	60 min	90 min	120 min	
**ad15**	147	110	112	107	104	−22.84
**ad16**	141	112	117	118	112	−21.71
Pioglitazone	139	105	110	112	115	−23.07

**Table 37 molecules-27-06763-t037:** Antidiabetic potential of the compounds **(ad17**–**ad18)** based on molecular docking studies.

Comp.	G Score	D Score	PMF Score	Chem Score	Interacting Residues
**ad17**	−273.19	−149.75	−81.45	−40.54	Tyr 327, His 449
**ad18**	−249.42	−162.88	−81.45	−40.54	Ser 289
Tesaglitazar	−279.20	−157.61	−67.78	−34.20	Ser 289, His 323, Tyr 473, Ser 342

**Table 38 molecules-27-06763-t038:** Hypoglycemic activity of the compounds (**ad19**–**ad20**) in streptozotocin-induced diabetic rats.

Comp.	% PPAR-γ Transactivation	Blood Glucose Level after 15 Days (mg/dL)
**ad19**	64.67	134.0 ± 5.09
**ad20**	63.78	139.6 ± 6.40
Pioglitazone	71.94	132.0 ± 5.20
Rosiglitazone	85.27	144.2 ± 6.12

All data are given as mean ± SEM analyzed by one-way analysis of variance (ANOVA), using Dunnett’s multiple comparison test applied for statistical analysis.

**Table 39 molecules-27-06763-t039:** Hypoglycemic activity of the compounds (**ad21**–**ad22**) using a streptozotocin-induced diabetic model in rats.

Comp.	Dose (mg/kg Body wt.)	% Reduction, Plasma Glucose (mg/dL) (mean ± SEM)
**ad21**	10/30/50	39.83 ± 0.29/44.62 ± 0.32/52.81 ± 0.32
**ad22**	10/30/50	36.76 ± 0.66/43.14 ± 0.35/49.99 ± 0.62
Rosiglitazone	10/30/50	38.57 ± 0.25/14.83 ± 0.18/12.74 ± 0.16

**Table 40 molecules-27-06763-t040:** PTP1B inhibition activity of the compounds (**ad23**–**ad24**).

Comp.	IC_50_(µM)
**ad23**	4.6
**ad24**	4.9
Ursolic acid	4.0

**Table 41 molecules-27-06763-t041:** Antidiabetic potential of the compounds (**ad25–ad27**) in alloxan-induced diabetes in rats.

**Comp.**	**Blood Glucose Level (mg/dL)**		
	**0 h**	**3 h**	**6 h**
**ad25**	343 ± 5.797	313.8 ± 9.411 ^**^	303.2 ± 9.827 ^***^
**ad26**	353.7 ± 6.026	315.8 ± 8.109 ^*^	311.2 ± 9.297 ^**^
**ad27**	341.5 ± 6.158	320.5 ± 6.737	313.3 ± 9.500 ^**^
Metformin	343.3 ± 6.206	322.8 ± 4.989 ^**^	292.0 ± 7.767 ^***^

(N = 5) All data are given as mean ± SEM, ^*^ *p* < 0.05, ^**^ *p* < 0.01, ^***^ *p* < 0.0001.

**Table 42 molecules-27-06763-t042:** Antidiabetic potential of the compounds (**ad28**–**ad30**) in alloxan-induced diabetes in rats.

Comp.	Mean ± SEM Blood Glucose Level (mg/dL)					
	0 h	1 h	2 h	4 h	6 h	8 h
**ad28**	305.3 ± 5.46 *	290.3 ± 7.32	200.3 ± 9.29	145.33 ± 1.76	102.0 ± 5.78 *	90.58 ± 4.73
**ad29**	306.0 ± 2.08	280.3 ± 3.85 **	208.3 ± 3.39	155.6 ± 3.48 **	110.3 ± 6.02	94.7 ± 4.41
**ad30**	316.0 ± 6.51 **	297.3 ± 6.37 *	195.3 ± 6.02	142.0 ± 8.67	105.3 ± 6.02 **	95.0 ± 2.89
Glibenclamide	383.8 ± 14.28	222.8 ± 8.05 **	180.3 ± 6.92	120.42 ± 9.86 *	93.6 ± 4.95	85.42 ± 2.53

All data are given as mean ± SEM analyzed by one-way analysis of variance (ANOVA), using Dunnett’s multiple comparison test applied for statistical analysis. ** *p* < 0.01 (considered as significant), * *p* < 0.001.

**Table 43 molecules-27-06763-t043:** Antidiabetic potential of compound **ad31** using PTP1B inhibitory studies.

Comp.	IC_50_ (µM)
**ad31**	9.96
Suramin	9.76

**Table 44 molecules-27-06763-t044:** Antidiabetic potential of the compounds **(ad32**–**ad33)** based on PTP1B inhibitory studies.

Comp.	IC_50_ (µM)
**40a**	6.89
**40b**	8.53
Suramin	9.76

**Table 45 molecules-27-06763-t045:** Antidiabetic potential of the compounds **(ad34**–**ad35)** in alloxan-induced diabetes in rats.

Comp.	% Decrease in Plasma Glucose Level (PG) at Various Drug Doses (mg/kg Bodyweight)		
	10 mg	30 mg	100 mg
**ad34**	42.48 ± 3.25	62.24 ± 3.42	70.35 ± 3.14
**ad35**	45.42 ± 1.25	58.36 ± 2.36	68.42 ± 2.16
Pioglitazone	47.25 ± 5.50	64.59 ± 5.42	75.43 ± 3.40

(N = 6) all data sare given as mean ± SEM value.

**Table 46 molecules-27-06763-t046:** Antidiabetic potential of the compounds (**ad36**–**ad39**) in streptozotocin-induced diabetes in rats.

Comp. (mg/kg Body wt.)	Blood Glucose Level (mg/dL)			
	0-Day	3-Day	7-Day	10-Day
**ad36**	87 ± 4.31	429 ± 7.77	246.66 ± 13.78	112 ± 10.13
**ad37**	80 ± 4.31	416 ± 7.77	240.66 ± 13.78	117 ± 10.13
**ad38**	85 ± 5.22	425 ± 9.34	233.52 ± 19.15	119 ± 18.54
**ad39**	82 ± 5.22	418 ± 9.34	232.52 ± 19.15	120 ± 18.54
Glibenclamide	89 ± 5.34	411 ± 13.11	227.45 ± 10.38	109 ± 13.16

(N = 10) All data are given as mean ± SEM analyzed by one-way analysis of variance (ANOVA) followed by the Tukey–Kramer multiple comparison test; *p* < 0.001.

**Table 47 molecules-27-06763-t047:** Antidiabetic potential of the compounds (**ad40**–**ad41**) using the alloxan-induced model in rats.

Comp.	Mean ± SEM Blood Glucose Level (mg/dL)					% Reduction in Blood Glucose Level after 14 Days
	0 h	3 h	6 h	24 h	14 days	
**ad40**	355 ± 24.59	322.8 ± 24.10	253.8 ± 23.45	231.4 ± 23.48	123 ± 18.7	65.35
**ad41**	376.4 ± 21.00	342.8 ± 21.58	315.2 ± 21.66	276 ± 21.79	146.4 ± 20.5	61.10
Metformin	441.8 ± 18.71	399.4 ± 17.72	289.4 ± 18.46	219.6 ± 18.40	112.8 ± 16.84	73.83
Pioglitazone	402.2 ± 28.7	363.4 ± 26.08	302.4 ± 26.87	232.2 ± 20.53	123.8 ± 16.94	69.22

**Table 48 molecules-27-06763-t048:** Antidiabetic potential of the compounds (**ad42**–**ad43**) in alloxan-induced diabetes in rats.

Comp. (mg/kg Body wt.)	Blood Glucose Level (mg/dL)			
	0-Day	3-Day	5-Day	7-Day
ad42	186.17 ± 1.16	198.23 ± 0.77	162.47 ± 1.22	109.45 ± 2.13
ad43	188.68 ± 1.23	195.35 ± 1.16	175.65 ± 0.86	118.63 ± 0.89
Rosiglitazone	188.45 ± 1.99	156.88 ± 0.82	125.77 ± 1.45	104.10 ± 1.72

(N = 6) All data are given as mean ± SEM analyzed by one-way analysis of variance (ANOVA) followed by Dunnett’s test.

**Table 49 molecules-27-06763-t049:** Antidiabetic potential of the compounds (**ad44–ad46**) based on α-amylase and α-glucosidase inhibitory studies.

Comp.	C log P (Lipophilicity)	MR (Molar Refractivity)	% Inhibition (250 µg/mL)	
			α-Amylase	α-Glucosidase
**ad44**	6.45	123	37.04	36
**ad45**	5.63	116.9	36.18	35.2
**ad46**	6.49	125.34	33.05	32.2
Acarbose	−6.6	141	43.05	40.91

**Table 50 molecules-27-06763-t050:** Antidiabetic potential of the compounds (**ad47**–**ad49**) using an alloxan-induced model in rats.

Comp.	Mean ± SEM Blood Glucose Level (mg/dL)					
	0 h	2 h	4 h	6 h	12 h	15th Day
**ad47**	259.33 ± 25.65	234.61 ± 17.45	209.33 ± 5.84	183.00 ± 3.21	161.65 ± 2.72	158.64 ± 2.90
**ad48**	256.69 ± 16.19	224.00 ± 7.63	198.33 ± 1.76	177.64 ± 2.02	159.00 ± 2.08	159.00 ± 1.15
**ad49**	253.00 ± 8.71	230.64 ± 4.33	207.65 ± 2.33	175.61 ± 3.93	162.34 ± 2.90	159.00 ± 0.57
Pioglitazone	310.64 ± 8.09	165.00 ± 3.60	145.67 ± 4.09	137.00 ± 2.64	119.33 ± 2.96	104.33 ± 2.33

(N = 6) All data are given as mean ± SEM analyzed by one-way analysis of variance (ANOVA) followed by Dunnett’s test.

**Table 51 molecules-27-06763-t051:** Antidiabetic potential of the compounds (**ad50**–**ad51**) according to α-amylase inhibitory activity.

Comp.	% Inhibition				IC_50_ (µg/mL)
	25 µg/mL	50 µg/mL	75 µg/mL	100 µg/mL	
ad50	32.59	51.78	66.98	81.30	22.35
ad51	27.77	53.23	62.27	79.43	27.63
Acarbose	37.35	53.45	73.25	88.57	21.44

**Table 52 molecules-27-06763-t052:** Antidiabetic potential of the compounds (**ad52**–**ad54**).

Comp.	Mean ± S.E.M Blood Glucose Level (mg/dL)
**ad52**	82.81 ± 1.115
**ad53**	86.31 ± 0.993
**ad54**	87.21 ± 1.233
Rosiglitazone	65.58 ± 1.013

**Table 53 molecules-27-06763-t053:** Antidiabetic potential of the compounds (**ad55**–**ad56**) using a sucrose-loaded model in rats.

Comp.	Blood Glucose Level (mg/dL)			
	0 h	1 h	2 h	4 h
**ad55**	96.5 ± 3.86	98.5 ± 5.01 ***	82.5 ± 2.87 ***	86.2 ± 4.25 ***
**ad56**	91.6 ± 2.17	91.5 ± 3.58 ***	86.2 ± 4.21 ***	80.6 ± 3.25 ***
Pioglitazone	98.5 ± 4.35	119 ± 3.57 ***	95.5 ± 5.12 ***	107.5 ± 3.62 ***

Each value is the mean ± S.D. for six rats, ***p < 0.001 compared with normal control. Data analyzed by using one-way ANOVA followed by t-test.

**Table 54 molecules-27-06763-t054:** In vitro antioxidant activity of the synthesized compounds (**ao1**–**ao2**) using the DPPH scavenging method.

Comp.	IC_50_ = µg/mL
**ao1**	09.18
**ao2**	12.67
Ascorbic Acid	40.00

**Table 55 molecules-27-06763-t055:** Antioxidant activity of the compounds **(ao3**–**ao4**).

Comp.	% Inhibition				IC_50_ = µg/mL
	25 µg/mL	50 µg/mL	75 µg/mL	100 µg/mL	
**ao3**	36.83	45.41	71.51	87.34	27.66
**ao4**	35.89	42.67	70.25	85.44	29.04
Ascorbic Acid	38.99	55.78	72.51	93.15	21.64

**Table 56 molecules-27-06763-t056:** In vitro antioxidant activity of the synthesized compounds (**ao5**–**ao6**) using the DPPH scavenging method.

Comp.	% Radical Scavenging Activity				IC_50_ = µg/mL
	25 µg/mL	50 µg/mL	75 µg/mL	100 µg/mL	
**ao5**	53.35	58.15	62.52	65.98	3.38
**ao6**	53.34	56.91	61.60	65.68	6.29
Ascorbic Acid	53.65	58.03	62.32	66.29	2.84

**Table 57 molecules-27-06763-t057:** Antioxidant screening results of the synthesized compounds (**ao7**–**ao8**) by applying the DPPH scavenging method.

Comp.	Mean abs ± SEM	% Inhibition
**ao7**	0.5616 ± 0.0005	66.80
**ao8**	0.5140 ± 0.0014	65.50
Ascorbic Acid	0.5260 ± 0.05	68.50

**Table 58 molecules-27-06763-t058:** Antioxidant screening results of the synthesized compounds (**ao9-ao10**) by applying the DPPH and ABTS^+^ scavenging methods.

Comp.	% Radical Scavenging	
	DPPH	ABTS
**ao9**	92.55	70.66
**ao10**	89.61	58.27
BHT	63.50	-
Trolox	73.62	54.35
Ascorbic Acid	77.20	-

**Table 59 molecules-27-06763-t059:** In vitro antioxidant activity of the synthesized compounds (**ao11**–**ao12**) using the DPPH scavenging method.

Comp.	% Radical Scavenging Activity					IC_50_ µg/mL
	10 µg/mL	50 µg/mL	100 µg/mL	250 µg/mL	500 µg/mL	
**ao11**	8.73	35.39	46.59	57.34	62.65	75.00
**ao12**	10.56	45.76	59.58	65.45	71.98	80.00
Ascorbic Acid	35.84	41.35	58.83	69.28	84.67	26.00

**Table 60 molecules-27-06763-t060:** Antioxidant activity of the synthesized compounds **(ao13-ao14**).

Comp.	EC_50_ (µg/mL)			
	DPPH Radical Scavenging Activity	Superoxide Anion Scavenging Activity	Lipid Peroxidation Inhibition	Erythrocyte Hemolysis Inhibition
**ao13**	58.68	79.94	131.79	96.62
**ao14**	52.18	101.18	149.70	107.28
Luteolin	44.18	31.01	149.70	107.28
Ascorbic Acid	40.28	21.01	139.97	96.63

**Table 61 molecules-27-06763-t061:** In vitro antioxidant activity of the synthesized compounds (**ao15**–**ao16**) using the DPPH scavenging method.

Comp.	% Inhibition				
	10 µg/mL	20 µg/mL	30 µg/mL	40 µg/mL	50 µg/mL
**ao15**	39.6	44.4	48.9	58.1	66.8
**ao16**	39.2	43.9	48.9	55.5	65.5
Ascorbic Acid	49.3	53.2	57.9	65.8	70.2

**Table 62 molecules-27-06763-t062:** In vitro antioxidant activity of the synthesized compounds **(ao17**–**ao18**).

Comp.	IC_50_ (µM)
**ao17**	940
**ao18**	998
Ascorbic acid	971

**Table 63 molecules-27-06763-t063:** In vitro antioxidant activity of the synthesized compounds (**ao19**–**ao20**).

Comp.	IC_50_ (µg/mL)
**ao19**	12.78
**ao20**	16.44
Ascorbic acid	23.15

**Table 64 molecules-27-06763-t064:** In vitro antioxidant activity of the synthesized compounds (**ao21**–**ao22**).

Comp.	IC_50_ (µg/mL)
**ao19**	10.78
**ao20**	11.16
Ascorbic acid	23.15

**Table 65 molecules-27-06763-t065:** Patent grant information of the thiazolidinedione analogues.

S. No.	Patent No.	Title	Procedure/Activity	Reference
1	WO/2019/016826	Novel 5-[4-(2-biphenyl-4-yl-2-oxo-ethoxy)-benzylidene]-thiazolidine-2,4-diones, their synthesis, and uses thereof.	Antidiabetic activity	[93]
2	WO/2002/026735	Sodium salts of 5-’4-’2-(n-methyl-n-(2-pyridyl)amino)ethoxy]benzyl]thiazolidine-2,4-dione.	Synthetic procedure	[94]
3	WO/2006/010345	Salt of oxalic acid with 5-[4-[2-(*n*-methyl-*n*-(2-pyridyl)-amino)ethoxy]benzyl]thiazolidin -2,4-dione and a method of its preparation and its use.	Antidiabetic activity	[95]
4	US 6,784,184 B2	5-(arylsulfonyl)-,5-(arylsulfinyl) and 5-(arylsulfanyl)-thiazolidine-2,4-diones useful for the inhibition of farnesyl-protein transferase.	Anticancer activity	[96]
5	WO/2003/053962	5-(4-(2-(n-methyl-n-(2-pyridyl)amino)ethoxy)benzyl)thiazolidine-2, 4-dione malic acid salt and its use against diabetes mellitus.	Antidiabetic activity	[97]
6	US 2020/0093812 A1	5-[[4-[2-[5-(1-Hydroxyethyl)pyridin-2-yl] ethoxy]phenyl]methyl]-1,3-thiazolidine-2,4-dione for treating nonalcoholic fatty liver disease.	Treatment of non-alcoholic fatty liver disease	[98]
7	EP0508740A1	Thiazolidinedione derivatives, their production, and their use.	Hypoglycemic and hypolipidemic activity	[99]
8	WO03053367A2	Hydrogenation of precursors to thiazolidinedione antihyperglycemics.	Anti-hyperglycemic activity	[100]
9	CN101531657A	Dimethyldiguanide of the thiazolidinedione pharmaceutical, preparation method and use thereof.	Anti-diabetic activity	[101]
10	CN104230915A	Thiazolidinedione-containing phenylpiperazine derivatives, as well as the preparation method and applications of thiazolidinedione-containing phenylpiperazine derivatives.	Anti-cancer activity	[102]
11	WO2010077101A2	Novel thiazolidinedione derivative and use thereof.	Treatment of CVS, renal disease, and GIT disease	[103]
12	JP2002322177A	Thiazolidinedione derivative.	Anticancer activity	[104]
13	WO2007109037A2	Thiazolidinedione analogues.	Antihypertensive activity	[105]
14	WO2017021634A1	Association between 3-[(3-{[4-(4-morpholinylmethyl)-1H-pyrrol-2-yl] methylene}-2-oxo-2,3-dihydro-1h-indol-5-yl)methyl]-1,3-thiazolidine-2,4-dione and a tyrosine kinase inhibitor of the EGFR.	Anticancer activity	[106]

## Data Availability

All data used to support the findings of this study are included within the article.

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
