# Peer review of "Thiazolidin-2,4-Dione Scaffold: An Insight into Recent Advances as Antimicrobial, Antioxidant, and Hypoglycemic Agents"

_molecules, 2022, doi:10.3390/molecules27196763_

Round 1

Reviewer 1 Report

1.  pay much attention to check the entire manuscript for English language corrections, with the help of a native English speaker.

2.  The plagiarism is a serious issue to be considered before any scientific publications.

3. The introduction, abstract and conclusion should be improved.

4. copyright permission should be added for all the figures and tables taken from other manuscript, review or book chapter.

5.The references should be updated

6. future prospects section should be added to the review

7. The abstract should be improved to clarify the aim of the work

8. More recent studies should be added to the review

9. In figure 13 the structure of compounds am 62, am63 were missed

10. Please add table of abbreviations

Author Response

Comment 1: pay much attention to check the entire manuscript for English language corrections, with the help of a native Authors agree with the suggestions of the reviewer and as suggested, the manuscript has been revised to refine the language. Comment 2: The plagiarism is a serious issue to be considered before any scientific publications. Yes, plagiarism is a serious issue, in our manuscript the data mainly consisted of IUPAC names of chemical moieties and their activity values, which is not accounted for plagiarism. Comment 3: The introduction, abstract and conclusion should be improved. Authors agree with the suggestion of the reviewer and said changes have been incorporated in the revised manuscript. Comment 4: copyright permission should be added for all the figures and tables taken from other manuscript, review or book chapter. Authors wants to clarify that all the figures and tables added in the manuscript are original and hence no copyright permission is required. Comment 5: The references should be updated Authors agree with the suggestions of the reviewer and necessary changes have been incorporated in the manuscript. Comment 6: future prospects section should be added to the review As suggested by the reviewer the future prospects section has been added along with conclusion in the manuscript. Comment 7: The abstract should be improved to clarify the aim of the work Authors thankful to the reviewer for improving the quality of the manuscript and as suggested the necessary changes have been made in the revised manuscript. Comment 8: More recent studies should be added to the review Authors agree with the suggestions of the reviewer and as suggested more recent studies has been added in the revised manuscript. Comment 9: In figure 13 the structure of compounds am 62, am63 were missed Authors appreciate Reviewer#1 for critically reviewing the manuscript. Figure 13 has been updated and necessary changes has been made in the revised manuscript. Comment 10: Please add table of abbreviations As suggested the table of abbreviations has been added in the revised manuscript.

Reviewer 2 Report

Comments and Suggestions for Authors

The manuscript entitled    " Thiazolidin-2,4-Dione Scaffold: An Insight into Recent Ad-2 vances as Antimicrobial, Antioxidant, Hypoglycemic Agents, 3 Mechanism of Action and Patents Granted " by Talha Bin Emran and etal, presented the pharmacological potential of TZD as antimicrobial, antioxidant, and  hypoglycemic agents along with their mechanism of action. This review paper will not only be helpful to researchers working on the development of new TZD analogs based on medicinal chemistry but also to de signing new drug molecules in future.

The manuscript is well-written and the finding is novel. Therefore it is appropriate that the manuscript can be accepted for publication in molecules after revision.

I have the following comments on the manuscript

-          Title of manuscript should be more concise

-          The rational of work should be more deeply explained

-          In its current state, the level of English throughout your manuscript does not meet the journal's desired standard. Please check the manuscript and refine the language carefully.

-          There are some typographical errors, it should be considered

Author Response

Comment 1:

Title of manuscript should be more concise.

Authors agree with the suggestions of the reviewer and hence has updated with following title.

Incorporated title:  Thiazolidin-2,4-Dione Scaffold: An Insight into Recent Advances as Antimicrobial, Antioxidant, Hypoglycemic Agents.

Comment 2:

The rational of work should be more deeply explained.

Authors agree with the comment of the reviewer. As suggested the authors tried to explain more about the rationale of work in the revised manuscript.

Comment 3:

In its current state, the level of English throughout your manuscript does not meet the journal’s desired standard. Please check the manuscript and refine the language carefully.

Authors agree with the suggestions of the reviewer and as suggested, the manuscript has been revised to refine the language.

Comment 4:

There are some typographical errors, it should be considered.

Authors appreciate the reviewer #2 for critically reviewing the manuscript and pointing out the typographical/grammatical errors in it. This will help to further improve the quality of the manuscript. As suggested, the manuscript has been revised thoroughly and necessary changes have been incorporated.

Round 2

Reviewer 1 Report

The revision done is sufficient, I recommended acceptance of the review in the present form